## RESEARCH ARTICLE

# LEAFY demonstrates functions in reproductive development of the gametophyte but not the sporophyte of the fern *Ceratopteris richardii*

Hannah McConnell[1], Jancee R. Lanclos[1], Katelynn Willis[1], Nicholas Gjording[1], Genevieve Stockmann[1], Catalina Lind[1], Julin N. Maloof[2], Andrew R. G. Plackett[3],* and Verónica S. Di Stilio[1],*

## ABSTRACT

Flowers are a key reproductive innovation of the angiosperms. Seed plant reproductive axes (including flowers) evolved as reproductively specialized shoots of the land plant diploid sporophyte, with the gamete-producing haploid gametophyte becoming reduced and enclosed within ovules and microsporangia. The transcription factor *LEAFY* (*LFY*) initiates floral development, yet it predates flowers and is found across all land plants. *LFY* function outside angiosperms is known from the moss *Physcomitrium patens*, where it controls the first cell division of the sporophyte, and from the model fern *Ceratopteris richardii*, a seedless vascular plant where *CrLFY1* and *CrLFY2* maintain vegetative meristem activity. However, how the floral role of *LFY* evolved remains unclear. Using overexpression, we uncover new roles for *CrLFY1* and *CrLFY2* in fern gametophyte reproduction, in sperm cells and in the gametophyte's multicellular notch meristem. While no sporophytic reproductive function was detected in terms of time to sporing, overexpression supports a role in frond compounding and in the first cell division of the zygote. Our findings suggest a potentially ancestral *LFY* function in fern haploid-stage reproduction, which might have been co-opted into the sporophyte during the origin of the flower.

KEY WORDS: Co-option, Fern, Notch meristem, Reproductive meristem, Sperm, Zygote

## INTRODUCTION

Flowers are a key innovation of angiosperms, the most recently diverging clade of land plants, that are largely credited with enabling one of the greatest evolutionary radiations of all time (Berendse and Scheffer, 2009). The primary role of the transcription factor *LEAFY* (*LFY*) is as a flower meristem identity gene, with loss-of-function mutants producing leaflike structures instead of flowers (Blázquez et al., 1997; Carpenter and Coen, 1990;

[1]University of Washington, Department of Biology, Seattle, WA 98195, USA.
[2]University of California, Department of Plant Biology, Davis, CA 95616, USA.
[3]University of Birmingham, School of Biosciences, Edgbaston, Birmingham B15 2TT, UK.

*Authors for correspondence (distilio@uw.edu; a.r.g.plackett@bham.ac.uk)

H.M., 0009-0004-2218-0089; J.R.L., 0009-0000-6405-7047; K.W., 0009-0004-9347-5845; N.G., 0009-0006-2253-9295; G.S., 0009-0004-8703-2327; C.L., 0009-0005-6896-7511; J.N.M., 0000-0002-9623-2599; A.R.G.P., 0000-0002-2321-7849; V.S.D.S., 0000-0002-6921-3018

Molinero-Rosales et al., 1999; Schultz and Haughn, 1991; Souer et al., 1998; Weigel et al., 1992), and constitutive expression resulting in early flowering (Weigel and Nilsson, 1995). *LEAFY* also falls into a special gene class known as pioneer transcription factors, a few of which are known in plants, with roles in developmental reprogramming (Lai et al., 2018; Yamaguchi, 2021). In its pioneer transcription factor role, *LFY* can bind DNA as either a monomer or dimer in heterochromatic regions, activating the expression of downstream genes by directly binding to promoters in heterochromatic regions and through the recruitment of chromatin-remodeling genes (Jin et al., 2021; Winter et al., 2011).

Even though flower meristem identity is considered to be the canonical role of *LFY*, its homologs are found across land plants, including those without flowers (Moyroud et al., 2010; Sayou et al., 2014), suggesting that its floral function evolved from a pre-existing ancestral role. *LEAFY* can also be active in vegetative shoot apical meristems of certain angiosperms (Kelly et al., 1995; Wang et al., 2008; Zhao et al., 2017; Shu et al., 2000; Moriyama et al., 2024; Souer et al., 1998) and gymnosperms (Mellerowicz et al., 1998; Mouradov et al., 1998; Shindo et al., 2001), the angiosperm sister group that produces seeds without flowers. Other functions for *LFY* homologs include regulating axillary meristems in rice (Rao et al., 2008) and compound leaf development in several angiosperms (Busch and Gleissberg, 2003; Champagne et al., 2007; He et al., 2020; Hofer et al., 1997; Jiao et al., 2019; Wang et al., 2013). In the model fern *Ceratopteris richardii* Brongn. (*C. richardii*), expression of a *LFY* paralog was found in developing fronds, and RNAi-mediated knockdown of its two *LFY* paralogs, *CrLFY1* and *CrLFY2*, identified roles in maintaining the vegetative meristem of the diploid sporophyte, in embryo development, and the apical cell (stem cell) of early-stage haploid gametophytes. Those results suggest that the vegetative meristem function of *LFY* was present in the most recent common ancestor (MRCA) of ferns and seed plants (Plackett et al., 2018). In the moss *Physcomitrium patens*, a bryophyte, expression of two *PpLFY* paralogs is found in the apical cell of the gametophyte, and their disruption results in the zygote failing to divide into a multicellular embryo, indicating that one or both paralogs are necessary for early sporophyte development (Tanahashi et al., 2005). The sporophyte shoot is believed to have evolved in the MRCA of all vascular plants through interpolation of an indeterminate vegetative apex into embryo development delaying the onset of spore production (Tomescu et al., 2014), with vegetative apices from all extant vascular plant lineages demonstrating core transcriptional similarities (Frank et al., 2015) and a defined transition from vegetative to reproductive development (Conway and Di Stilio, 2020; Plackett et al., 2014; Spencer et al., 2021; Zhao et al., 2025). The involvement of *LFY* in

regulating vegetative apices and cell divisions across land plants supports the hypothesis that this function is ancestral, whereas its role in floral meristem identity is derived, which thus suggests that this floral function arose through modification of the ancestral vegetative meristem function. However, there is also evidence in support of the role of *LFY* homologs in the reproduction of non-flowering plants. In seedless vascular plants, *LFY* expression has been detected in both vegetative and reproductive organs of the sporophyte (Himi et al., 2001; Rodríguez-Pelayo et al., 2022; Yang et al., 2017), and expression was also seen in reproductive structures (cones) of four conifer genera (Carlsbecker et al., 2004, 2013; Mellerowicz et al., 1998; Mouradov et al., 1998; Vázquez-Lobo et al., 2007). Whether *LFY* also has reproductive functions in *C. richardii* remains unclear, because transgenic knockdown of expression resulted in early termination phenotypes during vegetative development (Plackett et al., 2018). When heterologously expressed in an *Arabidopsis lfy* loss-of-function mutant, a gymnosperm *LFY* ortholog and the *C. richardii* paralog *CrLFY2* conferred a partial rescue of flower development, where some but not all floral whorls developed, while a moss version did not (Maizel et al., 2005). Although these experiments raise the hypothesis that *LFY*-dependent floral gene networks could have been co-opted from an ancestral *LFY*-dependent network still present in extant ferns, this remains to be tested once the native binding targets of *CrLFY2* are discovered.

The evolutionary history of *LFY*'s dual vegetative and reproductive meristematic roles thus presents a compelling question, particularly regarding the emergence of its highly specialized function in floral meristem development, motivating further research as the ability to perform functional studies continues to expand beyond seed plants (Di Stilio and Sinha, 2024). However, few *in planta* functional studies have investigated the potential reproductive role of *LFY* homologs outside the angiosperms. Ferns are representative of seedless vascular plants, sister to seed plants, and thus an excellent bridge group, with the homosporous fern *C. richardii* representing an effective model (Hickok et al., 1987, 1995; Hickok and Warne, 1998; Plackett et al., 2015; Renzaglia and Warne, 1995). In *Ceratopteris*, reproductive processes ontogenically equivalent to those of angiosperms occur in both the diploid shoot (sporophyte) and the haploid thallus (gametophyte). In the sporophyte, the shoot apex transitions from producing vegetative fronds to sporangium-bearing fronds (sporophylls), and these organs subsequently generate the haploid generation by meiosis, in the form of single-celled spores. Sexual reproduction occurs in the free-living multicellular haploid organism germinating from these spores, the thalloid gametophyte generating gametes (eggs and free-swimming sperm) in specialized organs (archegonia and antheridia) (Conway and Di Stilio, 2020). These gametangia arise from the activity of a 'notch' meristem – archegonia exclusively so (Banks, 1999). Typically described as a vegetative meristem due to its role in generating prothallus tissue, it has recently been shown that archegonia are specified behind the meristem in a position-dependent, cell lineage-independent manner (Geng, 2022), arguing that this meristem has reproductive functions analogous to the angiosperm inflorescence meristem. Unlike angiosperms, with their determinate microscopic, short-lived gametophytes, or bryophytes like *P. patens*, where the diploid sporophyte is short-lived, ferns exhibit free-living macroscopic and indeterminate haploid and diploid phases, making investigations into gene functions in both life stages feasible.

To test whether *LFY* exhibits reproductive activity outside the seed plants, i.e. in the development of gametangia and sporangia in

either the haploid or diploid stage, we characterized transgenic plants constitutively expressing *CrLFY1* and *CrLFY2* during the development of the fern *C. richardii*, bypassing the issue of developmental arrest at the early gametophyte stage caused by RNAi-mediated gene silencing (Plackett et al., 2018). Single and double over-expressors allowed us to investigate the possibility of sub- or neo-functionalization between the paralogs. On the one hand, our findings do not support the hypothesis of *CrLFY* playing a direct role in reproduction in the sporophyte phase (sporing), as expected from its known angiosperm role. On the other hand, we show new evidence of roles for *CrLFY* in gametophyte reproductive development, via the lack of sperm release from antheridia when downregulated and via the increased size of the notch meristem and gametophyte thallus when ectopically expressed. We also generated new functional evidence supporting a conserved role in the first division of the zygote, as previously described in moss (Tanahashi et al., 2005), with our data further suggesting that the expression of *CrLFY* must be spatiotemporally constrained to enable progression through the zygotic stage. Finally, our study further supports the role of *CrLFY* in the regulation of compound leaf development. Taken together, our findings provide the first evidence of a reproductive role for *LFY* in the *Ceratopteris* gametophyte, but not in the sporophyte. This raises the hypothesis of ancestral gametophytic reproductive functions for *LFY* in the , of ferns and angiosperms, from which *LFY*-dependent gene networks may have been co-opted into the sporophyte during the evolution of floral meristem identity, as sporophytes became the dominant phase in angiosperms.

## RESULTS

### Fern *LEAFY* misexpression affects frond development but not the sporing transition in sporophytes

Given that spore production in ferns can be considered analogous to flowering in angiosperms, and that *LFY* overexpression is known to advance flowering, we investigated the hypothesis that *CrLFY* overexpression advances sporing in *C. richardii*. To that end, we generated *35S::CrLFY1*, *35S::CrLFY2* and *35S::CrLFY1+2* transgenic lines and confirmed the presence of the constructs (Fig. S1A,B) and overexpression in the leaves (from here on 'fronds') of transgenic ferns before the emergence of sporophylls: the site of meiosis and sporogenesis (Fig. S1C-H). Compared to wild type, *35S::CrLFY1* sporophytes had 4.3- to 4.9-fold more expression of *CrLFY1* (Fig. S1C), gametophytes had 2.6- to 3.7-fold more expression (Fig. S1D), *35S::CrLFY2* sporophytes had 3.5- to 31-fold more expression of *CrLFY2* (Fig. S1E), gametophytes had 1.8- to 4.1-fold more expression (Fig. S1F) and the double mutant had approximately a fivefold expression of both paralogs (Fig. S1G,H). Although there was high variation in expression among transgenic plants, this variation did not directly correlate with phenotype. The number of inserts per transgenic line ranged from 2 to 12 (Fig. S2).

The time to reproductive transition did not vary between wild-type and transgenic genotypes, i.e. *CrLFY* overexpression did not accelerate sporing in either number of vegetative fronds produced prior to sporing or days post-fertilization to sporing (Fig. 1A,B, $n=20$, $P=0.72$ or $P=0.58$ two-way ANOVA). Transgenic plants occasionally produced an unusual number of fronds, either very few or many early in development (Fig. 1A), but nevertheless had, on average, the same number of vegetative fronds prior to sporophyll production and a similar variance compared to wild type ($F_{3,20}=3.21$, $P=0.072$). During sporophyll development, the number of sporangia produced per square cm did not differ between wild-type and transgenic fronds (Fig. S3A, $n=5$-10, $P=0.44$, two-way ANOVA), nor

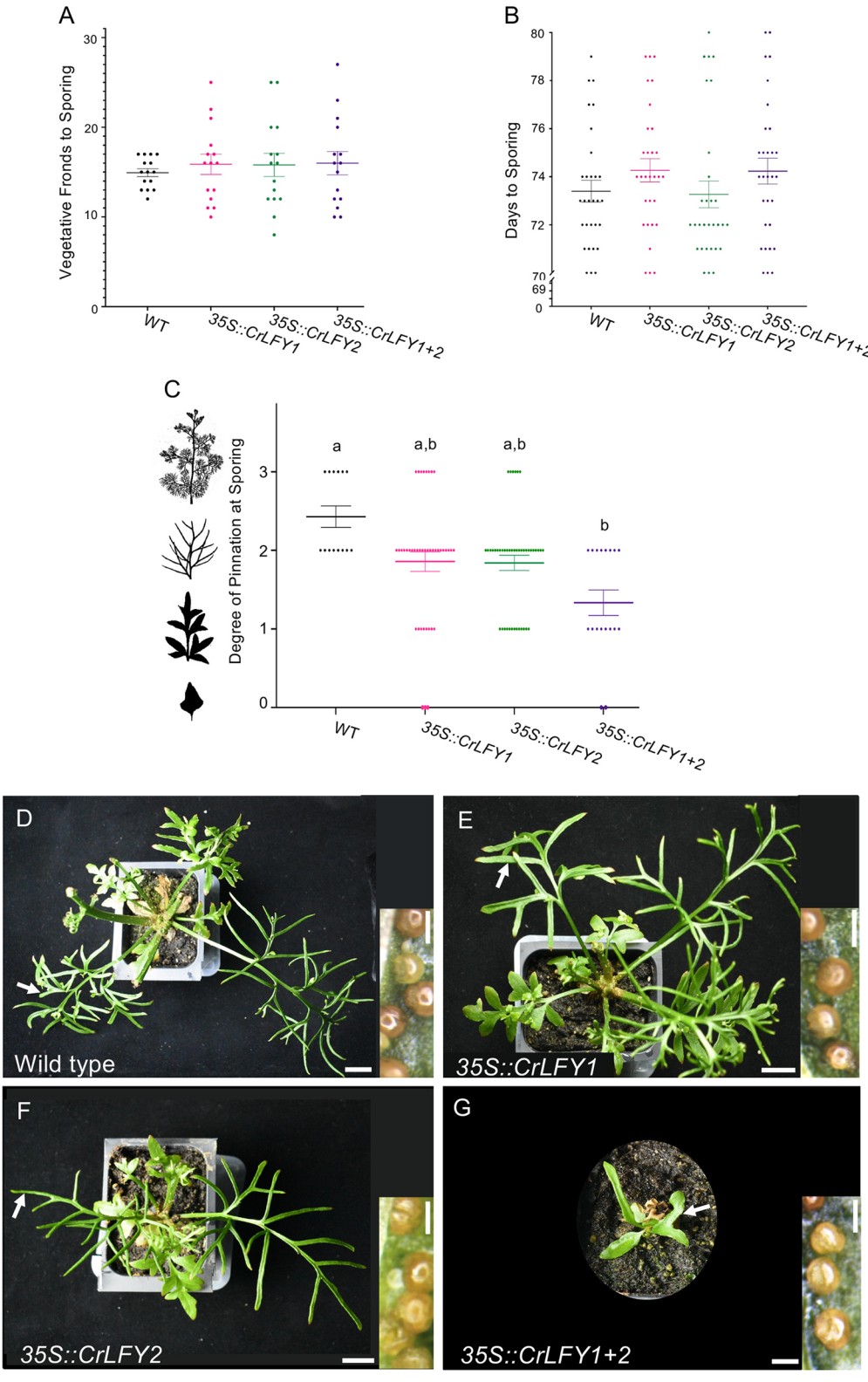

**Fig. 1. *LFY* orthologs affect frond development but not reproductive transition in *Ceratopteris richardii* sporophytes.** Overexpression of *CrLFY1* and *CrLFY2* does not change the time to sporing in sporophytes, resulting instead in abnormal development and simpler, less dissected sporophylls (spore-producing fronds) than in wild type at maturity. (A) Number of vegetative fronds produced before sporing. (B) Days to sporing for wild-type (WT) and transgenic plants (*n*=30, *P*=0.72, two-way ANOVA). (C) The degree of pinnation at sporing for wild-type and transgenic fronds: 0=simple or lobed, 1=pinnate, 2=bi-pinnate and 3=tri-pinnate, with representative frond silhouettes shown on the side (not to scale). Different letters denote a statistically significant difference (*n*=30, *P*<0.001, chi-squared test). Data are mean±s.e.m. (D-G) Sporophyll pinnation 72 days after fertilization (DAF). (D) Wild-type plant producing sporangia on the abaxial side of bi-and tri-pinnate sporophylls. (E) *35S::CrLFY1* and (F) *35S::CrLFY2* plants producing sporangia on pinnate and bi-pinnate fronds. (G) *35S::CrLFY1+2* plant producing sporangia on pinnate fronds. White arrows mark the sites where sporangia were photographed. Scale bars: 1 cm in main panels; 500 µm in insets. The dataset for this figure is Table S3.

did the length of sporangia-producing pinnae (Fig. S3B, n=10, P=0.99, two-way ANOVA). The germination rate of transgenic spores was lower at first, presumably due to the effect of antibiotics in the selective media, but was not significantly different from wild type by 180-200 h [Fig. S3C, n=3 (replicate plates, 50 spores each), P=0.46, two-way ANOVA]. Thus, there was no advancement in the sporing transition of ferns over-expressing *CrLFY*, contrary to predictions extrapolated from the role of *LFY* in angiosperms.

Like many ferns, wild-type *C. richardii* develops compound fronds that become increasingly dissected from simple, lobed, pinnate, bipinnate to tripinnate (Fig. 1C, frond silhouettes). Reproductive fronds (sporophylls) are bi- or tri-pinnate, and produce sporangia, the site of meiosis and sporogenesis (Conway and Di Stilio, 2020) (Fig. 1C). *LFY* orthologs have been shown to play a role in *C. richardii* frond development (Plackett et al., 2018) and leaf compounding in legumes (Hofer et al., 1997). In our experiment, the sporophylls of *35S::CrLFY1* and *35S::CrLFY2* plants ranged from simple to tri-pinnate (Fig. 1C,E,F), with no statistically significant difference from wild-type ratios (Fig. 1C,D, n=20-50, P=0.58, chi-squared test). In contrast, sporophytes overexpressing both *CrLFY* paralogs had sporophylls that were pinnate or bi-pinnate, but never tri-pinnate, and this was statistically significantly different from wild-type expectations (Fig. 1C,G, n=30, P<0.001, chi-squared test). Thus, fronds from double over-expressors displayed abnormal development, rather than the expected increased level of dissection from the described role of *LFY* orthologs in compound leaves of certain angiosperms.

## Misexpression of a fern *LEAFY* ortholog alters meristem size in the haploid gametophyte

Given the established role of *CrLFY* in the apical cell (a unicellular meristem) of the fern haploid stage (gametophyte), we tested the hypothesis that it may also be involved in regulating the multicellular 'notch' meristem of hermaphroditic gametophytes (whereas male gametophytes are lacking multicellular meristems; Banks, 1997). At 13 days post-sowing (dps), notch meristems of *35S::CrLFY2* and *35S::CrLFY1+2* sexually mature gametophytes both contained, on average, five additional meristematic cells compared to wild type (Fig. 2A,B,D-F, n=10, P<0.001). In contrast, notch meristems of *35S::CrLFY1* gametophytes were not significantly different from wild type (Fig. 2A-C,F, n=10, P=0.51), and native *CrLFY1* expression was not reported in the notch meristem of the previously published *CrLFY1pro::GUS* transgenic reporter line (Fig. S4; Plackett et al., 2018). Additionally, *35S::CrLFY1+2* gametophytes had, on average, 50.7% larger thalli compared to wild type, or gametophytes of the same age overexpressing either of the *CrLFY1* or *CrLFY2* paralogs (Fig. 2G, n=10, P<0.001). To test for native *CrLFY* function in the notch, meristem cell numbers were also quantified in *CrLFY*-RNAi lines at a comparable developmental stage, previously shown to knock down both paralogs (Plackett et al., 2018). One out of seven independent transgenic lines analyzed showed a statistically significant decrease in meristem cell number (Fig. S5, n=5, P<0.01); although there were highly variable expression levels between independent transgenic lines, this particular line was consistently knocked down (Plackett et al., 2018).

Because archegonia (egg-bearing organs) arise from the notch meristem, we quantified these and antheridia (sperm-bearing organs, which arise from mother cells at the periphery of the notch meristem; Banks, 1999) in *35S::CrLFY* hermaphrodites at 15 dps and 20 dps respectively, when notch-dependent processes could be expected to be even more affected. The average number of archegonia did not vary between genotypes (Fig. 2H, n=30, P=0.13, two-way ANOVA) but, interestingly, a small proportion of *35S:: CrLFY2* gametophytes (3/40 and 1/40, from two independent transgenic lines) failed to develop any archegonia, which was never

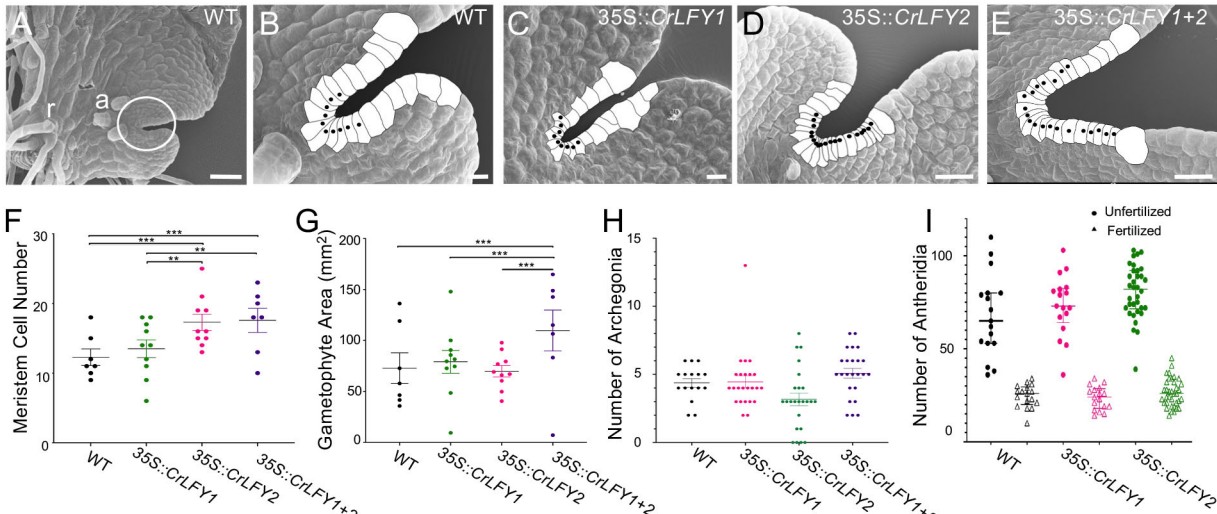

**Fig. 2. Misexpression of a *LEAFY* ortholog alters the multicellular meristem of the fern haploid stage.** *Ceratopteris richardii* gametophytes overexpressing *CrLFY2*, or both paralogs, produce more meristematic cells and are bigger, but with normal amounts of gametangia (antheridia and archegonia). (A-E) Scanning electron microscopy images before maturity (stage Gh6, Conway and Di Stilio, 2020) with notch area cells traced and filled in white, and black dots marking the presumed meristematic cells (with a length-to-width aspect ratio greater than 2:1). (A) Wild-type gametophyte, whole thallus (body) showing the notch area, circled and magnified in B, three archegonia (a) and rhizoids (r). (C-E) Notch meristem of representative transgenic gametophytes: (C) 35S::*CrLFY1*, (D) 35S::*CrLFY2* and (E) 35S::*CrLFY1+2*. Scale bars: 20 µm in A-C; 50 µm in D,E. (F) The number of meristematic cells in wild-type and transgenic gametophytes at 13 days post-sowing (dps, n=10, ***P<0.001, two-way ANOVA). (G) Gametophyte surface area in mm² (n=10, ***P<0.001, two-way ANOVA). (H) Number of archegonia in mature (15 dps) gametophytes (n=30, P=0.32, two-way ANOVA). (I) Number of antheridia at 20 dps in gametophytes that were either flooded at 15 dps or kept from fertilizing (n=20, P=0.90, one-way ANOVA). Data are mean±s.e.m. The dataset for this figure is Table S3.

observed in wild type (0/80). Wild-type and transgenic plants did not differ in the number of antheridia either, and all gametophytes produced more antheridia in the absence of fertilization regardless of genotype (Fig. 2I, *n*=20, *P*=0.56, one-way ANOVA). Together, these results represent the first evidence for a role of *CrLFY* in the multicellular notch meristem of *C. richardii* gametophytes, while also suggesting functional differentiation between the two paralogs.

### *CrLFY* genes are expressed in fern sperm cells

*CrLFY1* and *CrLFY2* expression had been previously detected in pooled gametophytes without distinction between the sexes

(Plackett et al., 2018). Here, we assessed the sex-specific expression of *CrLFY* in male and hermaphroditic gametophytes before and after sexual maturity. Both paralogs were significantly upregulated in sexually mature males at 14 dps (Fig. 3E), compared to immature males starting to undergo spermatogenesis at 8 dps (Fig. 3C), by 3-fold for *CrLFY1* and by 7-fold for *CrFLY2* (Fig. 3A,B, *n*=3, *P*<0.001, two-way ANOVA). Mature males were also significantly upregulated for *CrLFY* compared to immature hermaphrodites (Fig. 3D), by 6.8-fold for *CrLFY1* and by 7-fold for *CrLFY2*, and to mature hermaphrodites (Fig. 3F), by 12-fold for *CrLFY1* and by 27-fold for *CrLFY2* (Fig. 3A,B, *n*=3, *P*<0.001, two-way ANOVA). Hermaphrodites produce both archegonia and

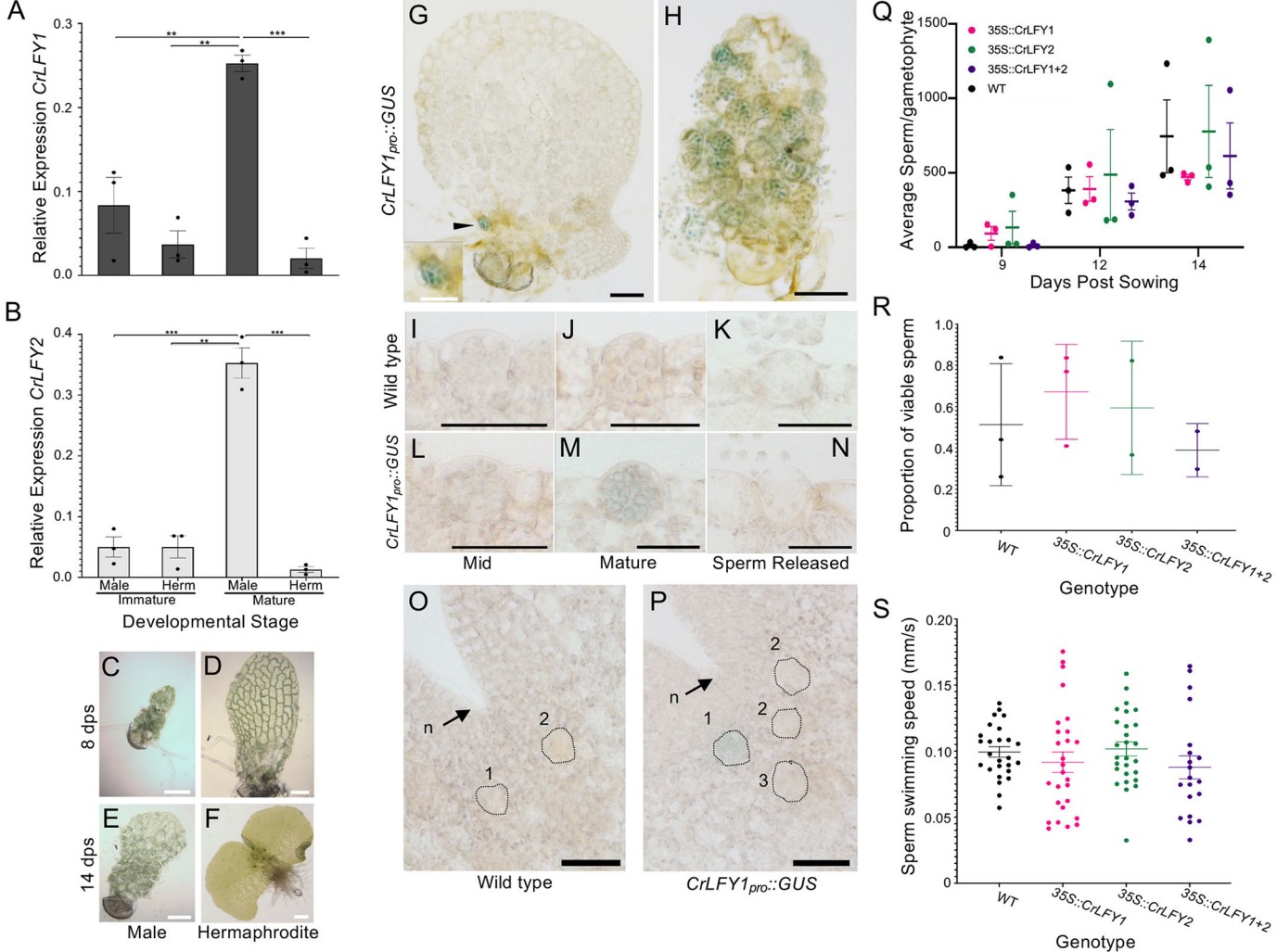

**Fig. 3. Fern *LFY* paralogs show high expression in mature sperm without a noticeable effect on sperm function when overexpressed.**
(A,B) Expression of two *C. richardii LFY* paralogs by qPCR (relative to the housekeeping genes *CrACT1* and *CrTBPb*) in gametophytes by sex (male and hermaphrodite, herm) and developmental stage (sexually immature and just starting to undergo spermatogenesis at 8 dps; mature at 15 dps). (A) *CrLFY1* and (B) *CrLFY2* (*n*=3, **P*<0.01, ***P*<0.001, two-way ANOVA). (C-F) Representative gametophyte images for the developmental stages used in qPCR: (C) immature wild-type (WT) male gametophyte at 8 dps, with few immature antheridia; (D) immature wild-type hermaphrodite gametophyte at 8 dps; (E) mature wild-type male gametophyte at 14 dps, covered in mature antheridia; and (F) mature wild-type hermaphrodite gametophyte at 14 dps, with antheridia and archegonia. Scale bars: 100 µm. (G,H) *CrLFY1_pro_::GUS* expression (blue) in (G) 9-day-old hermaphrodite and (H) male gametophytes. Scale bars: 100 µm; 50 µm (inset). (I-N) Developing antheridia in wild-type and (L-N) *CrLFY1_pro_::GUS* transgenic gametophytes undergoing spermatogenesis (I,L), with mature sperm (J,M) and after sperm release (K,N). Scale bars: 20 µm. (O,P) Archegonia developmental series from the notch meristem (n) in wild-type (O) and *CrLFY1_pro_::GUS* (P) gametophytes: (1) initiation, (2) development and (3) fully mature. Scale bars: 50 µm. (Q) The average number of sperm cells released from pools of 100 gametophytes at 9, 12 and 14 dps for wild-type and transgenic gametophytes (*n*=3, *P*=0.6432, two-way ANOVA). (R) The proportion of viable sperm (with propidium iodide as a viability stain) in wild-type and transgenic gametophytes (*n*=3 reps, 470 to 11,140 sperm analyzed per sample, *P*=0.51, mixed-effects binomial). (S) The swimming speed of sperm (in mm/s) for wild-type and transgenic gametophytes (*n*=25, *P*=0.4126, two-way ANOVA). Data are mean±s.e.m. The dataset for this figure is Table S3.

antheridia while males produce only antheridia, in high numbers. Thus, increased expression coincides with the increased number of antheridia containing fully developed sperm in mature male gametophytes. For *CrLFY1*, this result was supported by GUS localization specifically inside the sperm cells of the *CrLFY1$_{pro}$:: GUS* transgenic reporter line (Plackett et al., 2018) in both hermaphrodite (Fig. 3G) and male gametophytes (Fig. 3H). GUS expression was also detected in the apical cell of germinating spores carrying this reporter gene (Fig. S4), consistent with previously published *in situ* localization data (Plackett et al., 2018). Compared to wild-type controls (Fig. 3I-K), GUS staining was found in antheridia undergoing spermatogenesis (Fig. 3L), and in mature sperm (Fig. 3M), not in antheridia after sperm release (Fig. 3N). Compared to wild-type controls (Fig. 3O, two archegonia shown), GUS activity localized briefly to initiating archegonia (Fig. 3P, archegonium 1) and was not detectable throughout subsequent stages of archegonium development (Fig. 3P, archegonia 2 and 3).

### Fern *LFY* influences sperm release from antheridia

Because *CrLFY* is expressed in *C. richardii* sperm, we evaluated its putative role in sperm development and/or function. We investigated sperm performance parameters in *35S::CrLFY* transgenic gametophytes compared to wild type. There was no difference in the total number of sperm cells produced per gametophyte (Fig. 3Q, *n*=3, *P*=0.64, two-way ANOVA), in sperm viability (Fig. 3R, *n*=3, *P*=0.11, two-way ANOVA) or in sperm swimming speed (Fig. 3S, *n*=25, *P*=0.41, two-way ANOVA) between any transgenic line and wild type. Thus, *CrLFY* overexpression does not appear to influence sperm development or function.

In order to investigate a potential role of *CrLFY* in sperm development, we tested whether reduced *CrLFY* expression in *CrLFY* RNA interference (*ZmUbipro::CrLFY1/2-i3*) transgenic lines (Plackett et al., 2018) caused a sperm phenotype. We identified a significant decrease in the number of sperm released per gametophyte in *CrLFY*-RNAi gametophytes compared to wild-type plants at the mature developmental timepoints (Fig. 4A, *n*=3, *P*<0.0001, two-way ANOVA). At 9 dps, sperm were visible in antheridia in both sexes of wild type (Fig. 4B,D) and knockdown gametophytes (Fig. 4C,E). One day post flooding (dpf) wild-type sperm had been released from both sexes as expected from previous descriptions (Conway and Di Stilio, 2020) showing empty antheridia in both hermaphrodites (Fig. 4F) and males (Fig. 4H), while *CrLFY*-RNAi gametophytes still contained sperm within antheridia in hermaphrodites (Fig. 4G) and males (Fig. 4I). These results are consistent with *CrLFY* being required either for maturation of the sperm themselves or the surrounding antheridium.

### Misexpression of *C. richardii LEAFY* orthologs disrupts embryo development at the zygote stage

To further investigate the potential role of *CrLFY* in sperm development, we set up controlled fertilization assays in wild type, and in *35S::CrLFY1*, *35S::CrLFY2* and *35S::CrLFY1+2* transgenic lines, and recorded fertilization success as the emergence of visible embryos. Hermaphroditic gametophytes were isolated and flooded with sperm of the same genotype and followed for 14 dpf; by 10 dpf, 80% of wild-type gametophytes contained visible embryos, and by 14 dpf all showed visible signs of embryo development (Fig. 5A). Transgenic gametophytes for all three constructs had significantly fewer embryos from 10 dpf onwards

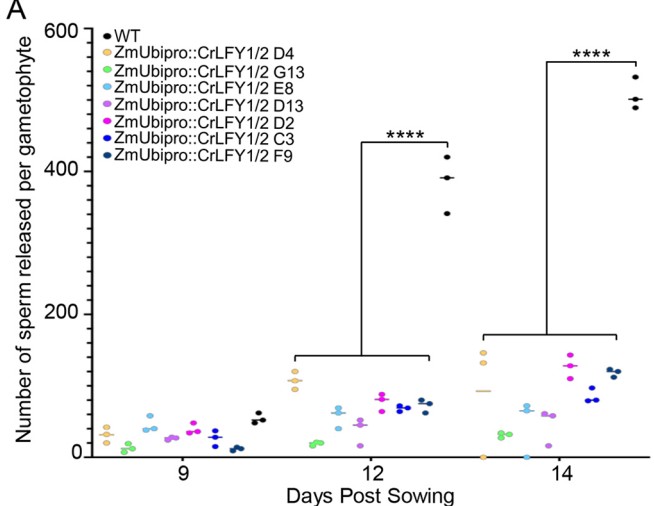

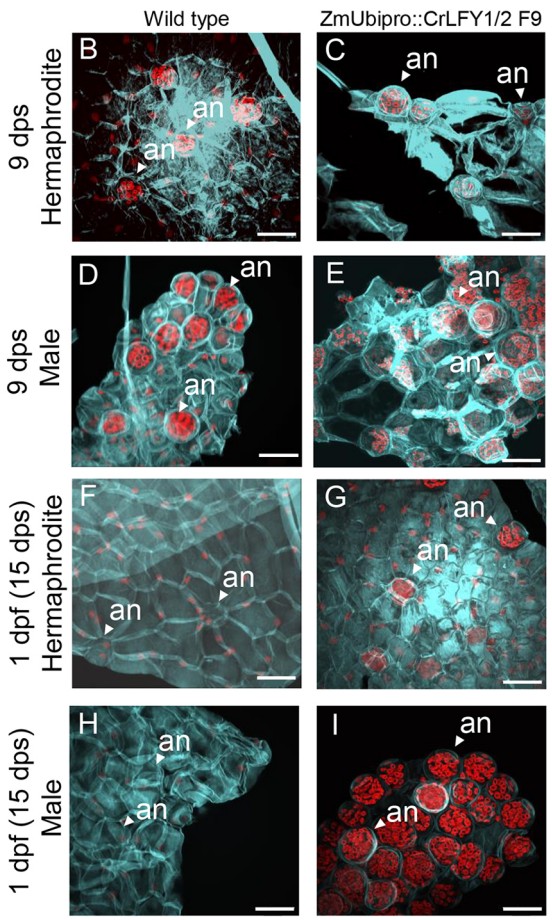

**Fig. 4. Gametophytes with reduced *CrLFY* expression fail to release sperm from antheridia.** (A) The average number of sperm cells released from pools of 100 gametophytes at 9 (immature), 12 and 14 (mature) days post-sowing (dps) for wild-type (WT) and seven independent *CrLFY*-RNAi transgenic lines (transgenic line naming as in Plackett et al., 2018) (*n*=3, ****P*<0.0001, two-way ANOVA). (B-I) Partial *z*-stacks from confocal images of representative wild-type and *CrLFY*-RNAi gametophytes at 9 dps; wild-type (B) and RNAi (C) hermaphrodite, wild-type (D) and RNAi (E) male. (F-I) Gametophytes one day post-flooding (dpf); wild-type (F) and RNAi (G) hermaphrodite, wild-type (H) and RNAi (I) male. All samples are stained with SR220 (light blue, marking the cell wall) and propidium iodide (red, marking the nuclei), showing multiple antheridia (an). Scale bars: 50 µm. The dataset for this figure is Table S3.

compared to wild type, with 36% of *35S::CrLFY1*, 31% for *35S:: CrLFY2* and 30% of *35S::CrLFY1+2* gametophytes containing visible embryos from 10 dpf (Fig. 5A and Table S1, *n*=36, *P*<0.01, two-way ANOVA). This result suggests that *CrLFY* overexpression disrupts either fertilization or post-fertilization processes.

To further determine whether the decrease in visible embryos was due to a maternal or a paternal effect, individual wild-type hermaphroditic gametophytes were isolated and flooded with transgenic sperm from each of the three genotypes, and the reciprocal experiment consisted of flooding transgenic gametophytes with wild-type sperm. In crosses where wild-type hermaphrodites were flooded with transgenic sperm, there was a significant reduction in the proportion of gametophytes with embryos after 10 dpf compared to the wild-type×wild-type control (Fig. 5B and Table S1, *n*=36, *P*<0.01, two-way ANOVA), with 46% of *35S::CrLFY1*, 48% of *35S::CrLFY2* and 44% of *35S::CrLFY1+2* hermaphrodite gametophytes containing visible embryos 10 dpf. Out of concern for the possibility of selfing of wild-type hermaphrodites with their own wild-type sperm, all

resulting sporophytes were genotyped (see Materials and Methods) and found to be transgenic. In contrast, when transgenic gametophytes were flooded with wild-type sperm, there was a significant reduction in the proportion of gametophytes with embryos between 10 and 13 dpf (*n*=36, *P*<0.01, two-way ANOVA), but by day 14 they experienced a rescue, whereby the proportion of visible embryos was no longer significantly different from wild-type (Fig. 5C and Table S1, *n*=36, *P*=0.06, two-way ANOVA). Additionally, the proportion of transgenic gametophytes fertilized with wild-type sperm containing visible embryos was significantly higher between 10 and 13 dpf compared to those fertilized with transgenic sperm, whether they were wild-type or transgenic (Table S1), with 56% for *35S::CrLFY1* sperm, 57% for *35S:: CrLFY2* sperm and 56% for *35S::CrLFY1+2* sperm (*n*=36, *P*<0.05, two-way ANOVA). Thus, the reduced proportion of embryos developing from transgenic fertilizations, along with the partial rescue by wild-type sperm, suggest that *CrLFY* misexpression affects early sporophyte development through paternal effects.

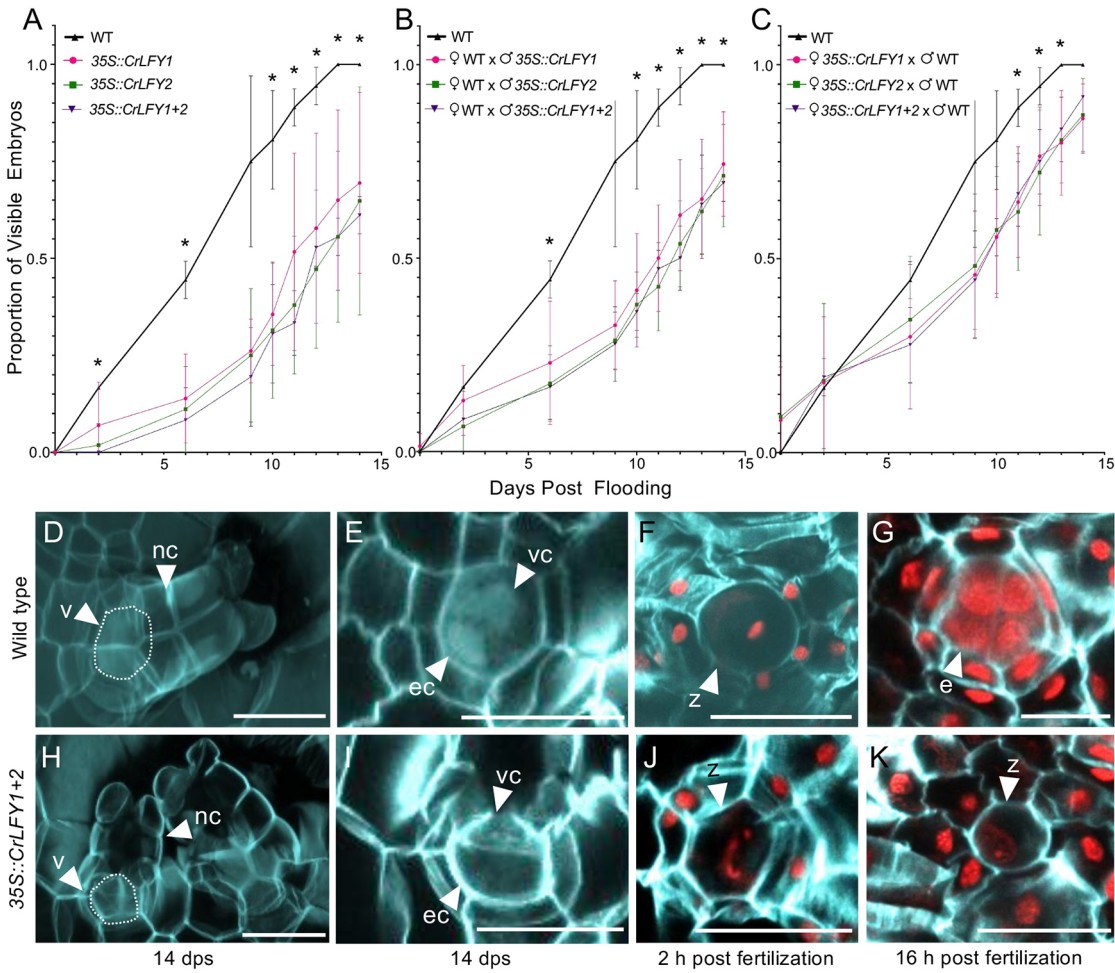

**Fig. 5. Misexpression of fern *LEAFY* homologs disrupts early development of the *C. richardii* embryo.** In controlled fertilization assays, overexpression of fern *LEAFY* orthologs *CrLFY1* and *CrLFY2* prevents zygotes from undergoing cell division and developing into multicellular embryos. (A-C) Time series of the proportion of gametophytes producing visible sporophytes after flooding (up to 14 dpf) with water containing sperm from the same genotype (A) or outcrossed (B,C). Data are mean±s.e.m. (*n*=36, *P*<0.05, see Table S2 for individual *P*-values). (D,H) Partial *z*-stacks from confocal images comprising the surface of an archegonium in wild-type (D) and *35S::CrLFY1+2* gametophytes before fertilization (H), stained with SR220 (light blue, cell wall) and Hoechst (light blue, nucleus), denoting the neck canal (nc) and venter (v) cells. White tracing indicates the location of the egg and ventral canal walls, not visible in the partial stack. (E,I) A single slice taken from the same *z*-stack, further zoomed in, denoting the ventral canal (vc) and egg cell (ec, white arrowheads) at the center of the venter, for wild-type (E) and transgenic (I) gametophytes. (F,J,K) A zygote (z) is shown at the center of the archegonial venter for wild-type (F) and transgenic (J,K) gametophytes 1 day after flooding, stained with SR2200 and propidium iodide (red nuclei). (G) A wild-type multicellular embryo (e) developing inside the archegonial venter 1 day after flooding. Scale bars: 20 µm. The dataset for this figure is Table S3.

To investigate the mechanism underlying the decrease in visible embryos under *CrLFY* misexpression, we examined early embryo development in the archegonia of wild-type and transgenic gametophytes after flooding. Wild-type mature gametophytes produced archegonia with neck cells and a venter containing a ventral canal cell and an egg cell (Fig. 5D,E, Movie 1), as expected from earlier descriptions (Lopez-Smith and Renzaglia, 2008), and *35S::CrLFY1+2* gametophytes showed similar morphology to these controls (Fig. 5H,I). After flooding, fertilized wild-type gametophytes produce a zygote inside the venter, morphologically distinguishable from the egg cell by the increased cell size, taking up the whole ventral canal space (Lopez-Smith and Renzaglia, 2008). Based on these guidelines, we observed that sperm successfully entered the cytoplasm of both wild-type and transgenic eggs by 2 h post-flooding (Fig. 5F,J, Movie 2). Within 16 h, 88% of the wild-type gametophytes (22 out of 25) contained a multicellular embryo (Table 1, Fig. 5G, Movie 3) and the rest contained an unfertilized egg cell (Movie 4), while only 60% of double transgenic gametophytes (*35S::CrLFY1+2*) contained embryos (15 out of 25) and the rest of the transgenic gametophytes exhibited a single cell with an elongated, more diffuse nucleus that we interpret as a zygote arrested before the first cell division and/or not having undergone proper syngamy (Table 1, Fig. 5J,K, Table 1, Movie 5). Occasionally, a wild-type gametophyte with an embryo also contained an arrested zygote from a second fertilization event in another archegonium (Fig. S6), and these had a single condensed nucleus. In gametophytes where *CrLFY1* and *CrLFY2* were downregulated by RNAi, 84% contained a multicellular embryo 16 h after fertilization, 8% had not been fertilized and still contained an egg cell, and 8% had been fertilized but their zygotes appeared to be arrested (Table 1); their nuclei were also spherical, as in wild type (Fig. S7), rather than elongated like the nuclei of zygotes overexpressing *CrLFY*.

*CrLFY1$_{pro}$::GUS* expression has previously been reported in the *Ceratopteris* embryo as early as the octant stage (Plackett et al., 2018) but earlier stages were not reported. Hence, we investigated the earliest timepoint after fertilization where native *CrLFY1* expression is present in this reporter line. We could not detect *CrLFY1$_{pro}$::GUS* staining in the zygote; GUS staining was first visible in the 2- to 4-cell embryo (Fig. S8). These expression results during early embryo development, together with the decrease in embryos when overexpressing *CrLFY* (Fig. 5A-C), suggest that the spatiotemporal expression of *CrLFY* requires precise regulation during zygote development. Misexpression of either *CrLFY* paralog in the zygote may prevent progression to the first cell division (by an unknown mechanism), causing developmental arrest of the young sporophyte.

**Table 1. Fertilization outcomes in wild-type and transgenic plants misexpressing *CrLFY***

| Genotype | Fertilization outcomes* | | |
| --- | --- | --- | --- |
| | Eggs | Zygotes | Multicellular embryos |
| Wild type | 0.12 | 0 | 0.88 |
| *35S::CrLFY1+2* | 0 | 0.4 | 0.6 |
| *ZmUbipro::CrLFY1/2-i3* | 0.08 | 0.08 | 0.84 |

*Proportion of eggs, zygotes and multicellular embryos in gametophytes 16 h post-fertilization (*n*=25) for wild-type and transgenic plants overexpressing the two fern *CrLFY* paralogs (*35S::CrLFY1+2*) or downregulated for both via RNAi targeted gene silencing (*ZmUbipro::CrLFY1/2-i3*). Arrested zygotes are found in overexpressing and RNAi transgenic lines, not in wild type.

## DISCUSSION

As the closest living relatives of seed plants, ferns provide a crucial link for understanding *LFY* function. Investigating *C. richardii* bridges the phylogenetic gap between angiosperms and bryophytes, offering a framework to trace the evolutionary trajectory of this master regulator of plant development. Here, we asked whether the sporophytic reproductive function of *LEAFY* in the angiosperms specifying floral meristem identity (Carpenter and Coen, 1990; Schultz and Haughn, 1991; Weigel et al., 1992) arose from an ancestral vegetative shoot meristem role or an ancestral reproductive function (in gametogenesis or sporogenesis) pre-dating the angiosperms.

Consistent with previous research indicating that at least one of the *CrLFY* genes is necessary to maintain apical meristems in the fern sporophyte (Plackett et al., 2018), our overexpression analysis shows that both *CrLFY* genes together regulate frond dissection in the sporophyll. Transgenic plants overexpressing both *CrLFY* genes developed less dissected fronds, with fewer pinnae and pinnules (the smallest subdivided segment of a frond), which arise from pinnae initial cells (Hill, 2001), suggesting that *CrLFY* genes are involved in regulating these apical (stem) cells and that their misexpression inhibits pinna initial identity or activity. Given that *LFY* is found mostly as a single copy gene, and that it can function as both a monomer and a dimer in *Arabidopsis* (Winter et al., 2011), it is unclear at this point whether the compound leaf phenotype is due to an additive effect of increased total *CrLFY* expression or to a regulatory interaction between the two paralogs that would suggest sub- or neo-functionalization. Interestingly, decreased leaf compounding in plants overexpressing *CrLFY* matches that of plants where *CrLFY* was downregulated by RNAi (Plackett et al., 2018), contrary to the predicted increase in dissection based on the role of *CrLFY* in promoting leaf apical cell divisions. This unexpected result suggests that the overall spatial and temporal *CrLFY* expression pattern is more crucial than the total expression level in determining pinnae and pinnule outgrowth. Moreover, proper maintenance of pinnae initial identity would require the absence of *CrLFY* expression in surrounding cells for leaf development to proceed normally, consistent with GUS expression patterns reported previously for *CrLFY1* (Plackett et al., 2018). Alternatively, it is also possible that *CrLFY* affects leaf development in a way other than by regulating cell division at pinnae initials. Several angiosperm *LFY* homologs promote compound leaf development, e.g. in legumes (Champagne et al., 2007; He et al., 2020; Hofer et al., 1997; Jiao et al., 2019; Wang et al., 2025), and this role in marginal meristems leading to compound leaves may have arisen separately in ferns and legumes, given the growing consensus that fronds have evolved independently from seed plant leaves (Tsuda, 2024). However, there is also evidence of *LFY* homologs regulating compound leaf development in California poppy, an early-diverging eudicot (Busch and Gleissberg, 2003), which, together with our findings, suggests the alternative hypothesis that this function could represent a deep homology present in the ancestor of ferns and seed plants.

Given that overexpression of *LFY* accelerates flowering in *Arabidopsis* (Weigel et al., 1992), the lack of acceleration of the sporing transition in our transgenic plants was unexpected. However, an increase in *CrLFY* expression from vegetative fronds to sporophylls is not naturally found in *C. richardii* (Plackett et al., 2018) and, consistent with this evidence, *CrLFY* overexpression did not alter the development of reproductive structures (sporangia and spores) in sporophylls. While early termination of the *Ceratopteris* vegetative shoot in loss-of-function experiments prevented addressing this question, the evidence shown here suggests that

the floral function of *LFY* did not arise from an ancestral role in promoting a reproductive transition in the fern sporophyte shoot. Other fern and lycophyte *LFY* orthologs show increased expression in sporophylls compared to vegetative fronds (Rodríguez-Pelayo et al., 2022); however, fern expression data did not distinguish between sporangia and sporophyll tissue, and the only available expression data specifically in sporangia are from heterosporous lycophytes (Rodríguez-Pelayo et al., 2022; Yang et al., 2017). Considering that *C. richardii* is a homosporous fern, a sporing transition role for *LFY* could either have been lost in the *Ceratopteris* lineage or gained in connection with the evolution of heterospory.

Our findings also support the capacity of *CrLFY2* to regulate the multicellular 'notch' meristem of hermaphrodite gametophytes. In addition to its floral meristem identity role, *LFY* orthologs are involved in the maintenance of shoot apical meristems in several dicot and monocot angiosperms (Kelly et al., 1995; Moriyama et al., 2024; Shu et al., 2000; Souer et al., 1998; Wang et al., 2008; Zhao et al., 2017), of axillary meristems in rice (Rao et al., 2008), and of marginal meristems in compound leaf development (already described). In gymnosperms, *LFY* paralogs are expressed in the vegetative shoot of *Gnetum* and *Pinus radiata* (Mellerowicz et al., 1998; Shindo et al., 2001), and either one or both *CrLFY* genes are necessary to maintain the apical cell in the early fern gametophyte (Plackett et al., 2018). However, this meristem function had not been previously shown in multicellular meristems of haploid gametophytes. Our overexpression experiments demonstrate that *CrLFY2* (but not *CrLFY1*) is capable of driving notch meristem proliferation. There is some additional support for this being a native function from the decrease in the number of notch meristem cells in a strong RNAi knockdown transgenic line. *CrLFY1_{pro}::GUS* expression is not reported in the notch meristem, supporting a paralog-specific role for *CrLFY2*. That only one of the two paralogs shows an effect when overexpressed suggests the possibility of sub- or neo-functionalization in regulating this meristem. Although the notch meristem is typically considered a vegetative meristem, it is specifically associated with the initiation of egg-producing gametangia (archegonia) at its periphery (Banks, 1999; Conway and Di Stilio, 2020; Geng et al., 2022; Hickok et al., 1987). Archegonium initiation has recently been demonstrated to be dependent on positional cues in relation to the notch meristem rather than cell lineage (Geng, 2022). This supports reclassification of the notch as a reproductive meristem, with functions analogous to the angiosperm inflorescence or floral meristem. CrLFY2 has been shown to be capable oft partially complementing an *Arabidopsis lfy* loss-of-function mutant (Maizel et al., 2005), suggesting its potential to perform angiosperm *LFY* function given the right genomic context. However, the native downstream targets of CrLFY2 have yet to be identified to confirm functional conservation. In additional support of the reproductive identity of the notch meristem, we found evidence of roles for *LFY* in fern gametangia development. While we did not find differences in the average number of antheridia and archegonia between wild-type and transgenic plants, we observed a small number of *35S::CrLFY2* transgenic plants without archegonia, and we also detected *CrLFY1pro::GUS* signal in early archegonium development. Thus, our evidence suggests that *CrLFY2* may play a reproductive function in the fern gametophyte.

Our findings further identified the previously unreported expression of *CrLFY* in fern sperm. We determined that, in the gametophyte generation, *CrLFY* genes are most highly expressed in mature male gametophytes, and that in both sexes, *CrLFY1*

expression is localized to developing sperm within antheridia. Moreover, sperm is withheld in the antheridia of RNAi knockdown plants, rather than being released in response to flooding from otherwise mature gametophytes, indicating a role for *CrLFY* genes in sperm maturation and/or release. *LFY* homolog expression has been reported in *Arabidopsis* stamens (containing pollen, the sperm-producing male gametophytes) (Nakabayashi et al., 2005; Schmid et al., 2005) and in antheridia of the bryophyte *Marchantia polymorpha* (Arnoux-Courseaux and Coudert, 2024; Kawamura et al., 2022), although not of moss (Tanahashi et al., 2005). Recent phylogenetic analyses have resolved the bryophytes as monophyletic, giving the evidence from these latter two lineages equal weighting (OTPTI, 2019). While functional genetic studies have not yet been performed to elucidate LFY function in these contexts, this male organ-associated expression suggests that the *Ceratopteris* sperm function identified is potentially ancestral in seedless plants.

Stalled zygote development in a small proportion of *CrLFY* knockdown gametophytes suggests that decreased expression of *CrLFY* might prevent cell division in the zygote. This role was previously hypothesized based on *CrLFY1pro::GUS* expression in the early embryo (Plackett et al., 2018), and the evidence presented here represents the first functional support for the role of *CrLFY* in promoting the transition of the zygote into a multicellular embryo. The presence of arrested zygotes in *PpLFY* disruptant mutants of the moss suggested that this is an ancestral role of *LFY* in the early development of the sporophyte (Tanahashi et al., 2005), and evidence presented here suggests that this function was retained in vascular plants and subsequently lost in angiosperms (or in seed plants).The first zygotic cell division in *Arabidopsis*, *Ceratopteris* and *Physcomitrium* is asymmetric (Gooh et al., 2015; Johnson and Renzaglia, 2008; Mansfield and Briarty, 1991; Tanahashi et al., 2005; Ueda et al., 2011; Wang et al., 2020), and analogous to animal embryology, there is evidence of morphometric gradients behind this asymmetry in *Arabidopsis* (Lukowitz et al., 2004; Wang et al., 2006). The molecular mechanisms underlying zygote polarity and the first asymmetric division in land plants remain unknown (Matsumoto and Ueda, 2024). *LFY* is not expressed or known to function in the *Arabidopsis* zygote and embryo, and hence this role has presumably been lost in angiosperms, although this aspect deserves further exploration in other angiosperms (Blázquez et al., 1997; Klepikova et al., 2016; Weigel et al., 1992). Native *CrLFY* expression is asymmetric in early multicellular fern embryos (Plackett et al., 2018), but GUS staining is insufficiently sensitive to detect possible intracellular gradients in late zygote development. Alternatively, *CrLFY* may influence or establish another native intracellular gradient necessary for the first division of the zygote. Auxin has long been suggested as a zygotic gradient (Zhang and Laux, 2011), and in *Arabidopsis* floral tissues, *LFY* is both regulated by and regulates auxin biosynthesis (Li et al., 2013), providing a potential mechanism to explain the developmental arrest observed.

Intriguingly, we also found that overexpression of *CrLFY* causes zygotic arrest in *C. richardii*. In gametophytes flooded with sperm overexpressing *CrLFY*, regardless of the genotype of the egg, more than half of the zygotes failed to form embryos. The fact that zygotic arrest occurred in similar proportions across *CrLFY1*- and *CrLFY2*-overexpressing genotypes suggests that both *CrLFYs* are capable of interfering with zygote development. We determined that zygote arrest through *CrLFY* overexpression occurred when it was introduced through the sperm and was associated with an abnormally elongated and uncondensed nucleus not seen in stalled *CrLFY-RNAi* zygotes, suggesting that development was affected at

a different stage. Given that we detected no native zygotic expression of *CrLFY1* between 4 and 24 h post-flooding, we speculate that *LFY* must first be repressed during fertilization to allow proper development of the zygote. It is unclear whether a similar repression of *LFY* homologs is necessary for zygote development in *Physcomitrium* or other bryophytes. The nuclear morphology observed could potentially represent incomplete syngamy. Zygotic genome activation (ZGA) is the transition of the zygote genome from silent to transcriptionally active (Fu et al., 2024), and while most evidence suggests a higher role for the maternal genome in this process, there is also evidence for a paternal contribution (Gehring et al., 2004; Kermicle, 1970; Pignatta et al., 2018). In mammals, pioneer transcription factors are heavily involved in ZGA (Fu et al., 2024; Kobayashi and Tachibana, 2021), and LFY has been described as a pioneer transcription factor in *Arabidopsis* (Jin et al., 2021; Yamaguchi, 2021). Further investigations will be necessary to determine whether *CrLFY* genes also behave as pioneer transcription factors that contribute to ZGA, in which case, their overexpression could advance the timing of ZGA and affect proper nuclear fusion. Our results thus raise the testable hypothesis that *CrLFY* gene expression is tightly controlled for zygote development to progress through to the first cell division and that any disruptions in this expression pattern, whether via gene silencing or overexpression, will result in developmental arrest. The ultimate mechanism for the involvement of *CrLFY* in the proper formation and division of the first cell in the sporophytic phase will certainly require further investigation.

In conclusion, while we do not find evidence for a reproductive role of *CrLFY* genes in the sporophyte of the fern *C. richardii* via advanced sporing, we show that *LFY* homologs are involved in a previously unreported gametophyte reproductive role regulating sperm development. Our evidence further supports a probable gametophyte role regulating reproductive notch multicellular meristem activity. Hence, taking an evolutionary lens that considers available data across land plants, we hypothesize that *LFY* homologs may have regulated multicellular meristems in both the haploid gametophyte and diploid sporophyte stages of the MRCA of non-seed vascular plants and seed plants. This meristematic function would have become exclusive to the sporophyte in seed plants as vascular plants became increasingly sporophyte dominant in their evolutionary history, whereas gametophytes became determinate and hence ameristic. Our findings further suggest that LFY's derived angiosperm floral meristem identity function may have evolved from a general vegetative meristem maintenance role in the last common ancestor of ferns and seed plants, raising the testable hypothesis that this function may have evolved through co-option of existing reproductive *LFY*-dependent gene networks from the gametophyte to the sporophyte phase. Moreover, the regulation of zygote development by *LFY* appears to be conserved through at least the non-seed vascular plants and may be more nuanced than previously thought.

## MATERIALS AND METHODS
### Plant materials and growth conditions
All experiments were conducted in wild-type *Ceratopteris richardii* Hn-*n* accession background (Hickok et al., 1995). Spores were surface-sterilized by a 10 min treatment with 10% hypochlorite and 0.1% Tween (Sigma-Aldrich) at room temperature, rinsed four times and then imbibed in sterile MilliQ water for 2-6 days in the dark to synchronize germination ('Dark Start'; Hickok and Warne, 1998). Spores were then sown onto C-fern media at pH 6 in 1% Difco Bacto agar (Carolina Biologicals) with 20 µg/ml of

Hygromycin B (Millipore Sigma) as the selective agent for transgenic plants, and grown in a Percival chamber at 28°C, 16 h light/8 h dark, 80 µmol/m$^2$/s fluorescent light under humidity domes. Once gametophytes were sexually mature (Gh7, Conway and Di Stilio, 2020), the plates were flooded with 1 ml of sterile water to induce fertilization and incubated until the resulting sporophytes had five or six leaves. Young sporophytes were transplanted to soil (Sunshine #4, Sun Gro Horticulture, Agawam, MA, USA) in 24-well plug trays kept in standing water in a Conviron Chamber (28°C, 70% humidity, 16 h light/8 h dark) under 80 µmol/m$^2$/s fluorescent lights. After ~6 weeks, plants were transferred to 10×10 cm pots and moved to the greenhouse, kept in standing water and fertilized every 2 weeks (Plant Marvel Nutriculture, 20-10-20+, 9.6 g/10 gal). Mature sporophylls were cut and placed in glassine bags to mature for at least 3 months, after which spores were collected into microcentrifuge tubes and stored at room temperature in the dark.

### Transgenic line preparation
#### Generation of constructs
The coding sequences (CDSs) of *CrLFY1* and *CrLFY2* have previously been validated by cloning (Himi et al., 2001; Plackett et al., 2018). No other *LFY* homologs were detectable by amino acid sequence homology within the *C. richardii* genome v2.1 (Marchant et al., 2022). Each CDS was amplified from wild-type Hn-n *C. richardii* cDNA by RT-PCR and cloned separately into the *35S::ocs* expression cassette of the pART7 cloning vector (Gleave, 1992) as an *Eco*RI-*Bam*HI restriction fragment. The *35S:: CrLFY1::ocs* and *35S::CrLFY2::ocs* constitutive expression cassettes were each cloned separately into the 'pBOMBER' *Ceratopteris* transformation vector, carrying a hygromycin resistance cassette driven by the Gateway *nos* promoter (Plackett et al., 2015), as a NotI-NotI restriction fragment.

#### Generation of transgenic lines
Transformation of 35S::*CrLFY1* and 35S::*CrLFY2* constructs into Hn-n *C. richardii* callus was performed by microparticle bombardment, as previously described (Plackett et al., 2015). Each construct was transformed separately. T$_0$ sporophyte shoots were regenerated from transformed tissue under 40 µg/ml hygromycin selection. Spores were collected from regenerated plants, and germinated to yield T$_1$ gametophytes, which were self-fertilized to produce T$_2$ sporophytes after growing on hygromycin and genotyping to confirm construct presence. Double over-expressing transgenic lines were obtained by crossing the two validated T$_2$ single over-expressing lines BA14 and BD5, flooding single isolated hermaphrodites overexpressing one paralog with sperm overexpressing the other paralog. The resulting sporophytes were grown on hygromycin plates until transferred to soil and genotyped to confirm the presence of both constructs.

### Validation of transgenic plants
Sporophyte leaves from selfed T$_1$ plants were collected at the simple leaf stage (S2; Conway and Di Stilio, 2020), flash frozen and ground, and genomic DNA was extracted from 10 mg of tissue using a modified CTAB method (Doyle and Doyle, 1987). To validate the presence of the transgene, PCR was performed with construct-specific primers and confirmed T$_2$ sporophytes were assessed for construct insert number by digital droplet PCR (ddPCR), with primers and probes designed on the same exon region as the qPCR primers for both genes (Table S2). Thermocycling conditions were determined following the BioRad ddPCR Application guide. After droplet generation and thermocycling, samples were transferred to a QX200 droplet reader (Bio-Rad). Droplet counts were analyzed with QuantaSoft version 1.7 (BioRad) with default or manually adjusted settings for threshold determination to distinguish positive and negative droplets. *CrLFY1* copy number was calculated using *CrLFY2* as the internal reference in each sample, and the reverse was carried out for *CrLFY2* copy number calculation, with the wild-type reference copy number set to 1 in both cases.

### Phenotyping of transgenic plants
Phenotyping of transgenic plants was carried out in the T2 generation. Isogenic lines were produced by isolating hermaphrodite gametophytes at the G4 developmental stage when hermaphrodites and males can be

distinguished (Conway and Di Stilio, 2020), in a 24-well plate and flooding them at stage G7 when they had developed mature gametangia. All transgenic lines were grown alongside wild-type Hn-n. Phenotype characterization included: spore germination success, quantification of gametangia, gametophyte notch meristem size and cell number throughout development, quantification of archegonia with zygotes or embryos after fertilization, number of pinnae at sporing, pinnae length, days to production of sporangia and number of sporangia in sporophytes. Statistical analyses included ANOVA for multiple comparisons, and $\chi^2$ goodness of fit test for the number of pinnae per sporophyll.

### Gene expression analysis

RNA was extracted from flash-frozen whole single leaves (50-100 mg) or entire young sporophytes using the Spectrum Total Plant RNA kit (Sigma-Aldrich). Primers for *CrLFY1*, *CrLFY2* and the housekeeping genes *Actin* and *TBP* were as previously designed (Plackett et al., 2018). Primer amplification efficiency was determined with a plasmid serial dilution using the slope of the linear regression line. Primer specificity was tested via melting curve analysis. qRT-PCR of three biological replicates and three technical replicates each were performed in a Bio-Rad CFX Connect with iTaq Universal SYBR Green Supermix. *CrLFY* expression was calculated using the $2^{-\Delta\Delta Ct}$ method (Livak and Schmittgen, 2001) normalized against the geometric mean of housekeeping gene expression (Hellemans et al., 2007). The standard deviation of the Ct values of each gene was calculated to ensure minimal variation in gene expression. Error bars represent the standard error of the mean (s.e.m.) for the $2^{-\Delta\Delta Ct}$ values. Relative expression values of *CrLFY1/2* were compared amongst genotypes by one-way ANOVA followed by Tukey's comparisons.

### GUS staining

The *CrLFY1$_{pro}$*::GUS transgenic reporter lines used here were previously established (Plackett et al., 2018). Whole gametophytes were stained for GUS as described by Plackett et al. (2014, 2015) using 1 mg/ml X-GlcA and 20 µM potassium ferricyanide to increase staining strength and specificity. GUS solution was infiltrated into tissue without pre-fixation using a gentle vacuum over 2 min and incubated for 16-24 h at 37°C in darkness. GUS-stained tissues were cleared by incubating in 70% ethanol solution at room temperature.

### Microscopy

Live gametophytes were photographed on agar using a Q-imaging MicroPublisher 3.3 RTV camera mounted on an Amscope dissecting microscope, or a Zeiss Axio Zoom. For fluorescent microscopy, tissue was fixed in FAA (4% formaldehyde, 5% acetic acid and 50% ethanol) at 4°C overnight. ClearSee (Kurihara et al., 2015) was used for clearing for 7-14 days after fixation. Plants were stained with SCRI Renaissance 2200, SR2200 (Renaissance Chemicals, Selby, North Yorkshire, UK) and either Propidium Iodide or Hoechst 33342 (Thermo Fisher). Images were obtained with a Nikon A1R HD25 confocal microscope. Frond and whole-plant photos were taken using a Nikon D3400 hand-held camera with a macro lens attachment. Sporangia quantification was performed by dissecting 2 cm regions of mature sporophylls to reveal sporangia. Images were minimally processed for brightness and contrast using ImageJ (Schindelin et al., 2012). GUS-stained tissue was imaged under Kohler illumination using a Leica DM500 microscope mounted with a GXCAM-U3PRO-6.3 digital camera (GT Vision, Newmarket, UK). GUS staining images were minimally processed for brightness and contrast in Adobe Photoshop 2022.

### Fertilization assays

Wild-type and *35S::CrLFY* spores were sown on C-fern media as described above. Once gametophytes had developed enough for males and hermaphrodites to be differentiated (stage Gh4, Conway and Di Stilio, 2020), individual hermaphrodites were isolated into 24-well plates, 36 plants per each wild-type or transgenic line, and the rest of the population was allowed to continue to grow. Once gametophytes were sexually mature (Gh7, Conway and Di Stilio, 2020), the original plates they had been collected from were flooded with 2 ml water, and 1 ml of post-flood water (containing sperm released from the gametophytes) was collected with a

pipette and used to flood the isolated plants. For the first assay, all plants were flooded with water containing sperm of the same genotype. In the second assay, transgenic sperm were used to flood wild-type gametophytes; in the third, wild-type sperm were used to flood transgenic gametophytes. In order to test whether sporophytes in the second assay using wild-type hermaphrodites were the products of the intended cross, rather than of selfing, all resulting sporophytes were genotyped to confirm the presence of the transgene using methods described in the validation section above. Plants were checked under a dissection microscope for evidence of embryos at 2, 6 and 9 days after flooding, then daily for 14 days. Differences in the proportion of visible embryos were tested by two-way ANOVA followed by Tukey's comparisons.

### Meristem quantification

Wild-type and 35S::*CrLFY* gametophytes were fixed in FAA overnight just before sexual maturity (Gh6, Conway and Di Stilio, 2020). After overnight fixation in FAA, gametophytes were dehydrated to 100% ethanol through an alcohol series, critical point dried and sputter coated with gold particles. Samples were imaged on a JEOL JSM-6010 Plus scanning electron microscope and images were analyzed in FIJI v2.10 (NIH, USA) to determine the length and width of cells in the notch meristem area. Hermaphrodite gametophytes transition by dividing from a single apical cell to a multicellular lateral notch meristem as they develop; because cells that have recently divided periclinally in the notch meristem are narrow and long, we used a 2:1 length-to-width ratio to define meristematic cells. Meristematic cells were counted and the total surface area of the gametophytes was determined in ImageJ. Differences in the number of meristematic cells, the size of gametophytes and the number of gametangia (antheridia and archegonia) were compared by two-way ANOVA followed by Tukey's comparisons.

### Sperm performance assays

#### Sperm number and viability

Sperm number was determined from samples acquired by a BD FACSymphony A3 Cell Analyzer (Becton, Dickinson and Company, Franklin Lakes, NJ) using FACSDiva v. 8.0. Sperm viability, defined as the proportion of live sperm, was determined from samples acquired by a BD Accuri C6 Plus (Becton, Dickinson and Company). Spermatocytes were stained with propidium iodide and detected with a blue laser of 488 nm. All analyses were conducted using FlowJo v10.9.0 Software (BD Life Sciences) and gated initially on singlet cells. To determine viability, cells were gated to split two distinct propidium iodide staining intensities, where intense staining indicates a non-viable cell.

#### Sperm swimming speed

200 male gametophytes were collected and placed into distilled water. The suspension was pipetted off and placed on a microscope slide with a cover slip. Video of moving sperm was recorded on a Samsung Galaxy A54, mounted on a Leica DM1000 LED compound scope at 10× magnification. Raw videos were processed by DVR-Scan 1.6 software to make a black-and-white mask of the sperm for tracking; these masks were processed by TrackR v0.1.2. Scaled coordinates and frame number (30 frames per second) were used to calculate the speed from frame to frame of the first 10 sperm in view 5 min after the initial release of the sperm. These frame-to-frame speeds were then averaged for each of the sperm. Speed differences were compared by two-way ANOVA followed by Tukey's comparisons.

#### Sperm count

100 male gametophytes were collected at differing degrees of maturity [immature (Gm4, Conway and Di Stilio, 2020), mature (Gm7, Conway and Di Stilio, 2020) and 2 days after reaching sexual maturity], placed in water and allowed to release sperm for 1 min. The suspension was then pipetted off and placed in phosphate-buffered saline. These samples were stored at 4°C for at least 2 weeks. Propidium iodide at a final concentration of 0.4 mg/ml was added 45 min before analysis on the BD FACSymphony (Becton, Dickinson and Company). Using a HTS plate reader (Bruker, Billerica, MA), samples were analyzed at a flow rate of 1 µl/s. The number of sperm cells was compared by two-way ANOVA followed by Tukey's comparisons.

## Sperm viability

The sperm viability assay was designed as described by Zhang et al. (1992). 100 male gametophytes were collected at sexual maturity (Gm7, Conway and Di Stilio, 2020), placed in distilled water and allowed to release sperm for 60 s. The suspension was then pipetted off and placed in MilliQ water. This suspension was filtered through 30 μm nylon mesh (MTC Bio, Sayreville, NJ). Propidium iodide at a final concentration of 0.17 mg/ml was added to the suspension. Spherotech Accucount Blank particles (Biocompare, San Francisco, CA) with a concentration of $10^6$ were added to a final concentration of 77,000. After 6 min from gametophytes first being placed in water, samples were analyzed using a BD Accuri C6 Plus (Becton, Dickinson and Company) at a flow rate of 35 µl per second and a core size of 16 µm. Differences in viability were compared by two-way ANOVA followed by Tukey's comparisons.

## Acknowledgements

The authors thank Stephanie Conway and Karen Renzaglia for guidance on *Ceratopteris*-specific microscopy; Jo Bui, Valerie Bentivegna, Maggie Fuqua and Matt Akamatsu for assistance with ddPCR; Wai Pang Chan and the Biology Microscopy facility (University of Washington); Matt Footer for technical and microscopy support; and Aurelio Silvestroni (Department of Laboratory Medicine and Pathology Flow Cytometry Core, University of Washington) for technical and experimental design assistance. Barbara Ambrose provided advice on *in situ* hybridization. H.M. thanks her dissertation committee members Jennifer Nemhauser, Takato Imaizumi and Rebecca Price, and Anthony Garcia and the Di Stilio lab group for support and feedback.

## Competing interests

The authors declare no competing or financial interests.

## Author contributions

Conceptualization: H.M., A.R.G.P., V.S.D.S.; Formal analysis: H.M., N.G., J.N.M.; Funding acquisition: J.N.M., V.S.D.S.; Investigation: H.M., J.R.L., K.W., N.G., G.S., C.L., A.R.G.P.; Methodology: H.M.; Project administration: J.R.L., N.G.; Supervision: H.M., J.R.L., V.S.D.S.; Validation: H.M., C.L.; Visualization: H.M., K.W., N.G., G.S., V.S.D.S.; Writing – original draft: H.M., A.R.G.P., V.S.D.S.; Writing – review & editing: H.M., J.N.M., A.R.G.P., V.S.D.S.

## Funding

This work was supported by the National Science Foundation [IOS-1920408 (Developmental Mechanisms) to V.S.D.S. and J.N.M.]; the University of Washington's Kruckeberg-Walker Award, Orians Award for Tropical Studies and Frye-Hotson-Rigg Fellowship to H.M.; the Botanical Society of America's Bill Dahl Graduate Student Research Award to H.M.; the University of Washington's Walter and Margaret Sargent Award to K.W.; the University of Washington Fyre-Hotson-Rigg Award to K.W., G.S., C.L. and N.G.; a Mary Gates Research Scholarship to C.L.; and a Royal Society University Research Fellowship (URF\R1\191326 to A.R.G.P.). Open Access funding provided by the University of Washington. Deposited in PMC for immediate release.

## Data and resource availability

All relevant data and details of resources can be found within the article and its supplementary information.

## Peer review history

The peer review history is available online at https://journals.biologists.com/dev/lookup/doi/10.1242/dev.204808.reviewer-comments.pdf

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
