## [Peer Review File · Development (Cambridge, England)]

LEAFY demonstrates functions in reproductive development of the gametophyte but not the sporophyte of the fern *Ceratopteris richardii*

Hannah McConnell, Jancee R. Lanclos, Katelynn Willis, Nicholas Gjording, Genevieve Stockmann, Catalina Lind, Julin N. Maloof, Andrew R.G. Plackett and Verónica S. Di Stilio
DOI: 10.1242/dev.204808

Editor: Dominique Bergmann

Review timeline

Original submission:	25 March 2025
Editorial decision:	30 April 2025
First revision received:	2 September 2025
Editorial decision:	4 October 2025
Second revision received:	7 November 2025
Accepted:	10 November 2025

Original submission

First decision letter

MS ID#: dev.204808

MS TITLE: LEAFY demonstrates functions in reproductive development of the gametophyte but not the sporophyte of the fern *Ceratopteris richardii*

AUTHORS: Hannah McConnell, Jancee R. Lanclos, Katelynn Willis, Nicholas Gjording, Genevieve Stockmann, Julin N. Maloof, Andrew R.G. Plackett and Verónica S. Di Stilio

Dear Dr Di Stilio,

I have now received all the referees' reports on the above manuscript, and have reached a decision. The referees' comments are appended below, or you can access them online: please go to:

The referees express considerable interest in your work, as you will see in the reviewer comments, there was enthusiasm for the questions and model system. But, there were major concerns about the conclusions drawn from the data presented and a substantial revision and reframing of your manuscript would be required before we can consider publication. If you are able to revise the manuscript along the lines suggested, which may involve further experiments, I will be happy receive a revised version of the manuscript. Your revised paper will be re-reviewed by one or more of the original referees, and acceptance of your manuscript will depend on your addressing satisfactorily the reviewers' major concerns. Please also note that Development will normally permit only one round of major revision. If it would be helpful, you are welcome to contact us to discuss your revision in greater detail. Please send us a point-by-point response indicating your plans for addressing the referees' comments, and we will look over this and provide further guidance. It is important to keep the conclusions in line with what data are provided, being especially careful of conclusions made from overexpression data.

Please attend to all of the reviewers' comments and ensure that you clearly highlight all changes made in the revised manuscript. Please avoid using 'Tracked changes' in Word files as these are lost in PDF conversion. I should be grateful if you would also provide a point-by-point response detailing

how you have dealt with the points raised by the reviewers in the 'Response to Reviewers' box. If you do not agree with any of their criticisms or suggestions please explain clearly why this is so.

Reviewer 1

Advance summary and potential significance to field

Here, McConnell et al carefully assess the phenotypes of transgenic fern (*Ceratopteris richardii*) lines overexpressing CrLFY1 and/or CrLFY2, and expressing a GUS reporter. There are some interesting data here, including on CrLFY1/2 divergence, and on which cells and organs are sensitive to high LFY dosage, and on how high LFY dosage affects cell proliferation. Although I think the topic of what floral genes do in gametophytes and in flower-free species is important, and the experiments here have been carefully carried out, many of the conclusions in this paper are not supported by the data.

Comments for the author

The authors' data indicate that (a) pluripotent stem cells in embryos, gametophytic meristems, and sporophytic leaves are sensitive to LFY dosage, where high levels of LFY homolog activity increases cell division in some contexts (gametophytic meristems) and suppresses cell division in others (embryos, potentially sporophytic leaves); and (b) CrLFY homologs are expressed in mature male gametophytes, and may be expressed in sperm. To accurately infer what a gene's native function might be, data from over-expression or GUS reporter lines is best paired with mutant data, expression data of the gene in its native context at the same scale that function is being inferred, or other orthogonal pieces of data. These data are largely missing from this paper, and therefore none of the authors' conclusions about native LFY function in *C. richardii* gametophytes or sporophytes are supported. Before publication, there is a need for either more experimental data, or for the authors to reinterpret their data and reconsider their conclusions.

Specific Issues:

1. The title is not well supported by the data. Many of the issues I outline here occur throughout the paper, especially in the discussion. Here, I will focus here on the title:
 - a. I think the 'ancestral reproductive function' in the title refers to (a) expression of GUS reporter in sperm, and (b) the lower number of embryos produced from transgenic lines. Since, as far as I and the authors know 'this is the first report of expression in sperm' (line 233), the sperm expression could be a *C. richardii* synapomorphy. Since *C. richardii* is not 'ancestral', but an extant plant, just as evolved as any angiosperm, the 'ancestral' referring to sperm expression is inappropriate.
 - b. The GUS staining in Fig. 3 is the only evidence for sperm CrLFY expression. It looks like what was selected as 'the native promoter' of CrLFY1 was based on distance of sequence upstream of the start codon (1.9 kb upstream) (Plackett et al, 2018). What evidence do the authors have that this is the entire promoter, that recapitulates native expression patterns? How can they be sure they have captured the entire set of regulatory elements in that 1.9 kb? Although I can appreciate that this is a challenging system to work in, ensuring that this expression is not an artifact of the GUS experiment is particularly important because the sperm expression is novel.
 - c. In addition, the lower number of embryos produced in transgenic lines likely results from disrupted early cell divisions (Fig. 4), which seems like a cell division phenotype rather than a 'reproductive' phenotype.
 - d. The title suggests that CrLFY homologs don't function in the sporophyte because CrLFY overexpression doesn't accelerate sporulation by measuring days to sporing, and sporangia appear normal (Fig. 1). I don't think days to sporing is the right comparison to premature flowering. I interpret the premature flowering cause by LFY overexpression to be due to the meristem transitioning from an inflorescence meristem to a flower meristem. The analogous measure in *C. richardii* would be the number of vegetative leaves produced before producing sporophylls, which looks much smaller in Fig. 1F, even though it might not be happening earlier in days (i.e. developmental time is the right metric, not solar time). Regardless, without mutant data, this result doesn't tell us much about native CrLFY function, but instead tells us that both vegetative and reproductive (sporophytic) leaf primordia in *C. richardii* are sensitive to CrLFY dosage.
2. The abstract states that flowers... 'evolved as a modification of the ancestral plant life cycle whereby the gamete-producing generation (gametophyte) became enclosed within the diploid,

spore-producing generation (sporophyte)'. This is incorrect. Gametophytes are also endosporic in gymnosperms, and lycophytes like Isoetes and Selaginella.

Reviewer 2

Advance summary and potential significance to field

The manuscript by McConnell et al builds on previous work knocking down LFY orthologs in *Ceratopteris* by overexpressing these genes to infer the evolution of LFY function in land plants. The analysis of both sporophyte and gametophyte development reveals multiple unexpected phenotypes including regulation of gametophyte meristem size, early embryo development and potentially fertilization. The authors also demonstrate unexpected expression of LFY in sperm, although it is unclear if there is a function in that cell.

Overall, this is a very clear presentation of the gain of function phenotypes conditioned by over expression of CrLFY orthologs. I was particularly impressed with the thorough phenotypic analyses performed. The goal of this study is clearly stated: to infer CrLFY function to help reconstruct LFY's functional evolution from bryophytes to seedless vascular plants and ultimately flowering plants where LFY function is well characterized. I agree with the authors that the phylogenetic position of *Ceratopteris* makes this a crucial system to explore the evolution of an important developmental regulator across angiosperms, and their data is informative to that end. While I have no major concerns about the reproducibility of the results, I am less enthusiastic about their broad conclusions regarding CrLFY function and evolution based on primarily overexpression phenotypes. Ectopic and increased expression levels caused by the 35S promoter may or may not reflect endogenous LFY function. Indeed, against expectation, several of the reported GOF phenotypes largely phenocopy the LOF phenotypes previously reported. While the authors present several plausible explanations for this mismatch, the conflicting LOF and GOF data gives me pause about the functional relevance of the GOF phenotypes in general. For those phenotypes where overexpression phenotypes are opposite the LOF a strong case can be made the CrLFY is both necessary and sufficient. However, for phenotypes that only emerge in a GOF context or where the GOF and LOF are similar, the interpretation is much more problematic. Thus I would strongly recommend more care in the conclusions drawn and their potential relevance for functional evolution of LFY in land plants. I detail these concerns and a few others with more detail below.

Major Concerns

1. Line 89-93: The differential/partial rescue of *Arabidopsis lfy* mutants with heterologous genes from ferns and mosses does not mean that "LFY homologs have the potential to perform reproductive functions". A gene's function depends on the cellular/biochemical background it is expressed in. Thus, CrLFY could partially rescue *Arabidopsis* flower development simply because its protein sequence is similar enough to substitute biochemically for AtLFY, even though that same function is not present in a *Ceratopteris* context. Similarly, the moss PpLFY may be incapable of rescuing *lfy* because its protein sequence is too diverged even though they may share an ancestral role in reproduction.
2. Line 191: "Together, these results represent the first evidence for a role of CrLFY in the multicellular notch meristem of *C. richardii* gametophytes... This conclusion is not justified based on GOF alone. You would need to see a corresponding decrease in the notch meristem in LOF mutants to be sure this represents the endogenous gene activity.
3. Line 237: the conclusion that OEx does not affect sperm function seems premature. It was not fully clear to me from the materials and methods what the marker of sperm viability was measuring - simply the ability of sperm to be stained by propidium iodide? Failure to stain would indeed indicate inviability, but PI staining alone is not sufficient to conclude that the sperm are fully viable. The reduced zygote formation with a clear paternal effect could well indicate that the sperm have a reduced capacity to fertilize.
4. The nuclear staining of the zygote in Fig 4J is shaped abnormally compared to wt in Fig 4F where the nucleus is more compact and spherical. I'm not an expert on nuclear dynamics of fern embryo development, but this seemed strange. Could it indicate a lack of syngamy or nuclear fusion in these cells (again pointing to a sperm defect)?
5. While there was no change in days to sporing, was there a change in the number of leaves produced before the transition to sporophylls? Based on Fig 1F, it seems like this plant has fewer leaves than the corresponding wt in 1C. Since leaf number is often used as a proxy for phase change

in plants, any change in leaf number before sporing may be informative about the ability of CrLFY to facilitate a phase transition. This may also be relevant to the lack of leaf dissection seen in the LFY overexpressing lines, which in wt is correlated with a reproductive transition. If true, the fact that days to sporing did not differ may indicate a plastochron phenotype.

5. Line 324: The reduced zygote cell division from CrLFY GOF does not indicate a conservation of function with *P. patens* where failure of the zygote to divide is based on LOF. If anything it suggests they have opposite phenotypes.

6. Line 382-386: While there is no apparent role in sporing for CrLFY OEx (although see caveat above about leaf number), this GOF phenotype may not be relevant to LFY functional evolution. The evidence from LFY LOF in *Ceratopteris* related to its role in sporing is hard to engage since strong sporophytic LOF phenotypes failed to maintain the shoot apex.

7. Line 424: The word "reproduction" here (and throughout the manuscript) is used fairly loosely. Reproduction is not a single phenomenon in ferns or angiosperms but a general term for multiple discreet developmental phenomena. In angiosperms LFY plays a role in the transition to flowering and FM specification, but not necessarily in other aspects of reproduction (e.g. embryo sac or egg production, nor pollen/sperm cell production). In ferns there may be a role in sperm cell function or the notch meristem, however this is a distinct aspect of reproduction from LFY's role in angiosperms. I just worry that applying the term "reproduction" across these distinct developmental processes implies a homology with does not exist.

8. Line 441-453: The "conservation" of function between fern and moss LFY in early embryo divisions is not based on LOF in both systems, and thus highly problematic.

9. Paragraph beginning on line 454: This paragraph is too speculative for my taste given the lack of LOF phenotypes supporting a CrLFY role in early zygote division. In particular, the discussion of zygote asymmetry seems premature. Early asymmetric divisions may result from positional cues emanating from the archegonia or other surrounding gametophytic tissues, or even the entry point of the sperm (constrained by the archegonium). To my knowledge the egg of *Ceratopteris* or any other land plant has never been shown to have an internal morphogenetic gradient as is common in animals.

Minor suggestions:

Abstract line 29: change "where they control" to "where it controls"

Line 131: is there a reason why CrLFY can be so much higher in the sporophyte (31-fold)?

Line 217: 14 dps?

Line 282: Delete "Second"? I didn't see a First or Third to make a series.

Line 346: Rather than bridges the "functional gap" perhaps bridges the "phylogenetic gap" is more accurate?

Reviewer 3

Advance summary and potential significance to field

The authors describe a reproductive function of the LFY homologs in the gametophyte of the fern *Ceratopteris*. The gametophytic role of LFY is not accessible genetically using RNAi because it results in a developmental arrest at the early gametophyte stage. By overexpressing each of the two paralogs of LFY alone or in combination under 35S, they demonstrated that additional doses of CrLFY do not affect sporophyte reproduction (i.e., the formation of spores; the double overexpressor lines exhibited defects (simplification?) in leaf development) but affect the development of the notch meristem, which produces the gamete-forming organs. Overexpression of LFY2 or both LFY paralogs resulted in more meristematic notch cells and larger gametophytes (~50% for the double overexpressor), with a small proportion of the LFY2 overexpressor plants failing to develop archegonia, suggesting functional divergence between the two paralogs. Both LFY paralogs are also upregulated in the mature sperm cells, and LFY1 promoter GUS reporter is active in the maturing antheridia and sperm cells. Nonetheless, LFY overexpression did not affect sperm development and function. Crossing and fertilization experiments demonstrated that overexpression of CrLFY prevents normal cell division of the zygote and results in sporophyte abortion, a role which appears conserved between ferns and mosses.

This is a well-framed study, with a lot of new data clarifying some long-standing speculations in the field. The results are reported clearly and interpreted with care. The discussion provides a broad overview and a lot of valuable context. I have only minor comments.

The functional divergence of LFY1 and LFY2 is intriguing, and I wonder if the results can be explained by the higher level of overexpression of LFY2 (up to 31-fold) compared to LFY1 (up to 3.7-fold). Could you comment on the correlation between levels of overexpression for each paralog and the overexpression phenotypes?

In Fig. 2, could you include how meristematic cells were defined (and marked by a dot in A-E) - 2:1 L/W ratio?

Typo in Line 131: CrLFY2

Line 376: Lotus in this case refers to a legume, not an early-diverging angiosperm, please rephrase.

Line 462: could you clarify what you mean by "have clearly evolved"?

First revision

Author response to reviewers' comments

Point-by-point response to Reviewer Comments:

We thank all reviewers for their helpful comments and have tried to address their concerns to the best of our ability. Line numbers refer to the revised document, except when specified as "original". A highlighted copy of the revised manuscript is also provided.

The following figures/supplements were added in this revision in response to the reviewer's requests: Fig. 1 (Panel A); Figure 4 (all new); Table 1 (new). New Supplemental figures: Figs. S4, S5, S7 and S8

Reviewer 1: SUMMARY OF THE ADVANCE MADE IN THIS PAPER AND ITS POTENTIAL SIGNIFICANCE TO THE FIELD

Here, McConnell et al carefully assess the phenotypes of transgenic fern (*Ceratopteris richardii*) lines overexpressing CrLFY1 and/or CrLFY2, and expressing a GUS reporter. There are some interesting data here, including on CrLFY1/2 divergence, and on which cells and organs are sensitive to high LFY dosage, and on how high LFY dosage affects cell proliferation. Although I think the topic of what floral genes do in gametophytes and in flower-free species is important, and the experiments here have been carefully carried out, many of the conclusions in this paper are not supported by the data.

SUGGESTIONS TO AUTHORS

The authors' data indicate that (a) pluripotent stem cells in embryos, gametophytic meristems, and sporophytic leaves are sensitive to LFY dosage, where high levels of LFY homolog activity increases cell division in some contexts (gametophytic meristems) and suppresses cell division in others (embryos, potentially sporophytic leaves); and (b) CrLFY homologs are expressed in mature male gametophytes, and may be expressed in sperm. To accurately infer what a gene's native function might be, data from over-expression or GUS reporter lines is best paired with mutant data, expression data of the gene in its native context at the same scale that function is being inferred, or other orthogonal pieces of data. These data are largely missing from this paper, and therefore none of the authors' conclusions about native LFY function in *C. richardii* gametophytes or sporophytes are supported. Before publication, there is a need for either more experimental data, or for the authors to reinterpret their data and reconsider their conclusions.

Specific Issues:

1. The title is not well supported by the data. Many of the issues I outline here occur

throughout the paper, especially in the discussion. Here, I will focus here on the title:

We are aware of the correct use of the term “ancestral” and do not find instances of this term being used to refer directly to C. richardii, which we agree would be incorrect usage, but rather to the reconstruction of ancestral function. The only place where it could lead to an incorrect interpretation is in the Summary statement, where it is used succinctly due to the word limit, so we removed it there. Functional data from organisms representative of the main lineages in the phylogeny of land plants can be used to reconstruct a gene’s ancestral function. LFY’s function is known from angiosperms, a bryophyte representative, and it is inferred from expression for lycophytes, ferns, and gymnosperms. Our title refers to the C. richardii results presented in this study bridging a gap in functional knowledge for LFY orthologs and contributing to the reconstruction of an ancestral function for LFY, and we therefore believe that the title is justified and would prefer to keep it.

a. I think the 'ancestral reproductive function' in the title refers to (a) expression of GUS reporter in sperm,

Yes, this is partly what we meant, based on GUS expression and the results from the reproductive assays suggesting a male effect. This was specified in lines 33-34, and 300-301 of the original submission (now lines 33-34, and 284-285). Additionally, we have further support for a reproductive role via LFY’s involvement in sperm development from the results of RNAi knockdown (KD) on sperm, where sperm is not released from antheridia compared to WT plants at the same developmental stage (lines 247-248).

and (b) the lower number of embryos produced from transgenic lines.

We provide further clarification, since this is not what we meant: we are not considering the zygote development effects as part of the reproductive function, but rather as an issue in the early development of the sporophyte; the zygotes are stalled sometime after sperm entry. We consider instead the other component of reproductive function to be the effect on the notch meristem of the gametophyte, previously described as a “female” meristem [Geng 2022, Banks 1999, Hickok 1986], and on sperm cell (described in (a)). The former aspect may be less obvious but was specified in the abstract (lines 33-34) and again at the end of the introduction (lines 112-114 original, 128-130 revised): “we show new evidence of a reproductive role for CrLFY in the gametophyte, via its effect on the notch meristem that produces gametangia, and from the detection of expression in sperm cells”.

Since, as far as I and the authors know 'this is the first report of expression in sperm' (line 233), the sperm expression could be a *C. richardii* synapomorphy. Since *C. richardii* is not 'ancestral', but an extant plant, just as evolved as any angiosperm, the 'ancestral' referring to sperm expression is inappropriate.

Lines 414-497: We cite evidence for LFY expression in Arabidopsis stamens, likely from the male gametophyte (pollen grain) that carries two sperm cells, there is also expression data from Marchantia in antheridia by RNAseq (not sperm-specific, but in male gametangia; Arnoux-Courseaux and Coudert, 2024; Kawamura et al., 2022), already cited in the original manuscript (Lines 429-430 of the original). We believe that these reports in other plants reduce the likelihood that LFY expression in sperm is specific to C. richardii.

b. The GUS staining in Fig. 3 is the only evidence for sperm CrLFY expression. It looks like what was selected as 'the native promoter' of CrLFY1 was based on distance of sequence upstream of the start codon (1.9 kb upstream) (Plackett et al, 2018). What evidence do the authors have that this is the entire promoter, that recapitulates native expression patterns? How can they be sure they have captured the entire set of regulatory elements in that 1.9 kb? Although I can appreciate that this is a challenging system to work in, ensuring that this expression is not an artifact of the GUS experiment is particularly important because the sperm expression is novel.

Past empirical evidence shows knockdown phenotypes corresponding to expression patterns predicted by this promoter fragment from Plackett, Conway et al (2018) <https://elifesciences.org/articles/39625>, including expression in shoot apices corresponding to failure of shoot apex function in KD plants. Failure of gametophyte development and in situ hybridization evidence (Plackett, Conway et al., 2018) also corresponds to expression of GUS in the apical cell added to the supplements of this paper (Fig. S4); expression in embryo development (Plackett, Conway et al., 2018) corresponds to failure in embryo development in RNAi KD lines we have added into this manuscript (Fig. S7, Table 1), and expression in pinnae initials (Plackett, Conway et al., 2018) additionally corresponds to failure of frond compounding we show in this manuscript (Fig. 1).

c. In addition, the lower number of embryos produced in transgenic lines likely results from disrupted early cell divisions (Fig. 4), which seems like a cell division phenotype rather than a 'reproductive' phenotype.

The reviewer is correct, we did not intend to refer to this phenotype as reproductive, nor here, nor in the title (as noted above, 1b). We have now more specifically explained what we mean by "reproductive function" (Lines 104-106) early on since this was also requested by Reviewer 2.

d. The title suggests that CrLFY homologs don't function in the sporophyte because CrLFY overexpression doesn't accelerate sporulation by measuring days to sporing, and sporangia appear normal (Fig. 1). I don't think days to sporing is the right comparison to premature flowering. I interpret the premature flowering cause by LFY overexpression to be due to the meristem transitioning from an inflorescence meristem to a flower meristem. The analogous measure in *C. richardii* would be the number of vegetative leaves produced before producing sporophylls, which looks much smaller in Fig. 1F, even though it might not be happening earlier in days (i.e. developmental time is the right metric, not solar time).

Our title is about the reproductive function of LFY, which is based on its effects on gametogenesis and sporogenesis. We do agree that vegetative leaves to sporing is an appropriate measure of developmental time, and have added a graph to Fig. 1 (panel A) on vegetative fronds to sporing, which does not show statistically significant differences compared to WT. Therefore, based on our findings, we believe that this title holds true to finding a role of LFY in fern gametophyte reproduction, but not in the sporophyte.

Regardless, without mutant data, this result doesn't tell us much about native CrLFY function, but instead tells us that both vegetative and reproductive (sporophytic) leaf primordia in *C. richardii* are sensitive to CrLFY dosage

*Mutant data on the effect of CrLFY on fronds is published (Plackett, Conway et al, 2018), where we showed that RNAi affects frond development so that fronds do not properly progress to becoming sporophylls. Because of this limitation with the strong RNAi lines, we resorted to over-expression, which continues to support CrLFYs role in leaf development via its function in cell division in pinnae initials. Our experiments did not uncover earlier sporing as predicted from its angiosperm function. As overexpression of LFY in *Arabidopsis* induces early flowering, we also believe that it is within reason to assume that if CrLFY were involved in sporing, we would see a similar phenotype in these overexpressing plants, although we cannot entirely rule out a sporophytic reproductive role.*

2. The abstract states that flowers... 'evolved as a modification of the ancestral plant life cycle whereby the gamete-producing generation (gametophyte) became enclosed within the diploid, spore-producing generation (sporophyte)'.

This is incorrect. Gametophytes are also endosporic in gymnosperms, and lycophytes like *Isoetes* and *Selaginella*.

We are not referring here to heterospory and the endosporic gametophytes that have

independently evolved in the heterosporous ferns and lycophytes but, more generally, to the tendency towards a reduction of the gametophyte over evolutionary time, ultimately, all seed plant gametophytes are small and endosporic. We believe that in this context, the phrase may stand as written.

Reviewer 2: The manuscript by McConnell et al builds on previous work knocking down LFY orthologs in Ceratopteris by overexpressing these genes to infer the evolution of LFY function in land plants. The analysis of both sporophyte and gametophyte development reveals multiple unexpected phenotypes including regulation of gametophyte meristem size, early embryo development and potentially fertilization. The authors also demonstrate unexpected expression of LFY in sperm, although it is unclear if there is a function in that cell. Overall, this is a very clear presentation of the gain of function phenotypes conditioned by over expression of CrLFY orthologs. I was particularly impressed with the thorough phenotypic analyses performed. The goal of this study is clearly stated: to infer CrLFY function to help reconstruct LFY's functional evolution from bryophytes to seedless vascular plants and ultimately flowering plants where LFY function is well characterized. I agree with the authors that the phylogenetic position of Ceratopteris makes this a crucial system to explore the evolution of an important developmental regulator across angiosperms, and their data is informative to that end. While I have no major concerns about the reproducibility of the results, I am less enthusiastic about their broad conclusions regarding CrLFY function and evolution based on primarily overexpression phenotypes. Ectopic and increased expression levels caused by the 35S promoter may or may not reflect endogenous LFY function. Indeed, against expectation, several of the reported GOF phenotypes largely phenocopy the LOF phenotypes previously reported. While the authors present several plausible explanations for this mismatch, the conflicting LOF and GOF data gives me pause about the functional relevance of the GOF phenotypes in general. For those phenotypes where overexpression phenotypes are opposite the LOF a strong case can be made the CrLFY is both necessary and sufficient. However, for phenotypes that only emerge in a GOF context or where the GOF and LOF are similar, the interpretation is much more problematic. Thus I would strongly recommend more care in the conclusions drawn and their potential relevance for functional evolution of LFY in land plants. I detail these concerns and a few others with more detail below.

Major Concerns

1. Line 89-93: The differential/partial rescue of Arabidopsis lfy mutants with heterologous genes from ferns and mosses does not mean that "LFY homologs have the potential to perform reproductive functions". A gene's function depends on the cellular/biochemical background it is expressed in. Thus, CrLFY could partially rescue Arabidopsis flower development simply because its protein sequence is similar enough to substitute biochemically for AtLFY, even though that same function is not present in a Ceratopteris context. Similarly, the moss PpLFY may be incapable of rescuing lfy because its protein sequence is too diverged even though they may share an ancestral role in reproduction.

We have reworded the text in lines 89-92 to indicate that the cited study suggests the possibility of other LFY homologs performing reproductive functions, not a certainty that they do. The interpretation provided is from the authors of the cited study Maizel (2005) not our own, although we agree with it.

2. Line 191: "Together, these results represent the first evidence for a role of CrLFY in the multicellular notch meristem of *C. richardii* gametophytes..." This conclusion is not justified based on GOF alone. You would need to see a corresponding decrease in the notch meristem in LOF mutants to be sure this represents the endogenous gene activity.

We have added measurements of meristem cell numbers from CrLFY RNAi knockdown plants to the supplements (Fig. S5) indicating that there is a decrease in the number of cells in the meristem of strong CrLFY knockdown gametophytes, although we do not see a decrease in weaker knockdowns (lines 195-201).

3. Line 237: the conclusion that OEx does not affect sperm function seems premature. It was not fully clear to me from the materials and methods what the marker of sperm viability was

measuring - simply the ability of sperm to be stained by propidium iodide? Failure to stain would indeed indicate inviability, but PI staining alone is not sufficient to conclude that the sperm are fully viable. The reduced zygote formation with a clear paternal effect could well indicate that the sperm have a reduced capacity to fertilize.

Sperm "viability" is a term typically used to refer to the proportion of live sperm, although viability does not account for sperm quality, which is what we believe the reviewer is referring to. We have specified our use of the term viable in the methods in line 618. We additionally note that, in regard to sperm quality, the ability of the sperm to successfully fertilize suggests that there is no difference between the overexpressing sperm and WT, since we don't see a reduction in zygote number, just in embryos, although we acknowledge that we do not directly test for this. The sperm viability assay we use was based on Zhang et al., 1992 (<https://academic.oup.com/plphys/article/99/1/54/6088214>). Additionally, while we did not find an effect on sperm function due to overexpression, we have also investigated sperm in previously published RNAi KD lines of CrLFY, and we did find a developmental effect in these gametophytes, consisting of an inability of sperm to exit antheridia (lines 247-248).

4. The nuclear staining of the zygote in Fig 4J is shaped abnormally compared to wt in Fig 4F where the nucleus is more compact and spherical. I'm not an expert on nuclear dynamics of fern embryo development, but this seemed strange. Could it indicate a lack of syngamy or nuclear fusion in these cells (again pointing to a sperm defect)?

Thank you for noticing. This is an important point that we now address more fully in the revised text and related supplements. We also found this quite interesting and believe it may be indicative of a failure of complete nuclear fusion - there appears to be fusion of the nuclei to a point, but perhaps not full condensation of the nuclei; we now address this in the results in lines 300-302, and in the discussion in lines 454-555.

5. While there was no change in days to sporing, was there a change in the number of leaves produced before the transition to sporophylls? Based on Fig 1F, it seems like this plant has fewer leaves than the corresponding wt in 1C. Since leaf number is often used as a proxy for phase change in plants, any change in leaf number before sporing may be informative about the ability of CrLFY to facilitate a phase transition. This may also be relevant to the lack of leaf dissection seen in the LFY overexpressing lines, which in wt is correlated with a reproductive transition. If true, the fact that days to sporing did not differ may indicate a plastochron phenotype.

We acknowledge that including leaves to sporing is an important measurement of phase/developmental changes in plants and have included a graph on the number of leaves produced before the transition to sporophylls (Fig. 1 A) in addition to the days to sporing graph. These data show no difference in the number of fronds to sporing in overexpressing plants compared to WT.

6. Line 324: The reduced zygote cell division from CrLFY GOF does not indicate a conservation of function with *P. patens* where failure of the zygote to divide is based on LOF. If anything it suggests they have opposite phenotypes.

We have refined this argument both considering that we had not fully described our initial comparison, which was intended to indicate that CrLFY misexpression, regardless of LOF or OE, alters successful fertilization, possibly due to changes in expression patterns. We have also added investigations into CrLFY KD plants (new Fig. 4 and Fig. S7) and have since refined our hypothesis in the discussion based on the results of these investigations, discussing these two separate phenotypes starting on lines 419 and 443.

7. Line 382-386: While there is no apparent role in sporing for CrLFY OEx (although see caveat above about leaf number), this GOF phenotype may not be relevant to LFY functional evolution. The evidence from LFY LOF in *Ceratopteris* related to its role in sporing is hard to engage since strong sporophytic LOF phenotypes failed to maintain the shoot apex.

In Arabidopsis, GOF phenotypes are associated with early flowering; while we cannot entirely rule out a sporophytic reproductive role, it is reasonable to expect that if there is a GOF phenotype in Arabidopsis, we would also see one with GOF in Ceratopteris consisting of advanced sporing.

8. Line 424: The word "reproduction" here (and throughout the manuscript) is used fairly loosely. Reproduction is not a single phenomenon in ferns or angiosperms but a general term for multiple discreet developmental phenomena. In angiosperms LFY plays a role in the transition to flowering and FM specification, but not necessarily in other aspects of reproduction (e.g. embryo sac or egg production, nor pollen/sperm cell production). In ferns there may be a role in sperm cell function or the notch meristem, however this is a distinct aspect of reproduction from LFY's role in angiosperms. I just worry that applying the term "reproduction" across these distinct developmental processes implies a homology with does not exist.

We have made edits to define more strictly what we consider to be reproductive - we consider a reproductive role to be involved in the development of gametangia and sporangia and clarify that in lines 104-106.

89 Line 441-453: The "conservation" of function between fern and moss LFY in early embryo divisions is not based on LOF in both systems, and thus highly problematic.

We have refined this argument both considering that we had not fully described our initial comparison, which was that CrLFY misexpression, regardless of LOF or OE, alters successful fertilization, possibly due to changes in expression patterns. We have also added in investigations into CrLFY KD plants, and have since refined our hypothesis, separating out the GOF and LOF phenotypes in the discussion based on the results of these investigations, discussing them in sections starting on lines 419 and 443.

9. Paragraph beginning on line 454: This paragraph is too speculative for my taste given the lack of LOF phenotypes supporting a CrLFY role in early zygote division. In particular, the discussion of zygote asymmetry seems premature. Early asymmetric divisions may result from positional cues emanating from the or other surrounding gametophytic tissues, or even the entry point of the sperm (constrained by the archegonium). To my knowledge the egg of Ceratopteris or any other land plant has never been shown to have an internal morphogenetic gradient as is common in animals.

Neither Ceratopteris eggs nor zygotes have been shown to have an internal morphogenetic gradient, but a gradient does occur in Arabidopsis zygotes (Lukowitz et al., 2004; Wang et al., 2006), and it is possible that this occurs outside of the angiosperms. We have included this information to support our speculation in the text in lines 429-430. We have also since investigated zygotic cell division in CrLFY RNAi knockdown plants, and address the resulting phenotypes in the text in a paragraph starting at line 419, suggesting that zygotic stalling of CrLFY RNAi plants may be due to a loss of morphometric gradients, but that the stalling caused by CrLFY OE may be due to a different mechanism.

Minor suggestion

Abstract line 29: change "where they control" to "where it controls": ***It has been changed.***

Line 131: is there a reason why CrLFY can be so much higher in the sporophyte (31-fold)?

We cannot ascertain why the CrLFY paralogs are much more highly expressed in the sporophyte, but we hypothesize that the levels of CrLFY are more tightly regulated in the gametophyte. We have previously published work using RNAi knockdowns of CrLFY (Plackett, Conway et al 2018), and the degree of knockdown is much stronger in the sporophyte than in the gametophyte, suggesting that there are mechanisms in the gametophyte regulating CrLFY expression that are not at play in the sporophyte.

Line 217: 14 dps? ***This was a typo, changed to 14 dps instead of 15.***

Line 282: Delete "Second"? I didn't see a First or Third to make a series.

There was a "first" a few lines above; we have decided to remove them as they were quite spaced out.

Line 346: Rather than bridges the "functional gap" perhaps bridges the "phylogenetic gap" is more accurate? ***Great suggestion, thank you, this has been changed in the text (line 328).***

Reviewer 3: The authors describe a reproductive function of the LFY homologs in the gametophyte of the fern *Ceratopteris*. The gametophytic role of LFY is not accessible genetically using RNAi because it results in a developmental arrest at the early gametophyte stage. By overexpressing each of the two paralogs of LFY alone or in combination under 35S, they demonstrated that additional doses of CrLFY do not affect sporophyte reproduction (i.e., the formation of spores; the double overexpressor lines exhibited defects (simplification?) in leaf development) but affect the development of the notch meristem, which produces the gamete-forming organs. Overexpression of LFY2 or both LFY paralogs resulted in more meristematic notch cells and larger gametophytes (~50% for the double overexpressor), with a small proportion of the LFY2 overexpressor plants failing to develop archegonia, suggesting functional divergence between the two paralogs. Both LFY paralogs are also upregulated in the mature sperm cells, and LFY1 promoter GUS reporter is active in the maturing antheridia and sperm cells. Nonetheless, LFY overexpression did not affect sperm development and function. Crossing and fertilization experiments demonstrated that overexpression of CrLFY prevents normal cell division of the zygote and results in sporophyte abortion, a role which appears conserved between ferns and mosses.

This is a well-framed study, with a lot of new data clarifying some long-standing speculations in the field. The results are reported clearly and interpreted with care. The discussion provides a broad overview and a lot of valuable context. I have only minor comments.

The functional divergence of LFY1 and LFY2 is intriguing, and I wonder if the results can be explained by the higher level of overexpression of LFY2 (up to 31-fold) compared to LFY1 (up to 3.7-fold). Could you comment on the correlation between levels of overexpression for each paralog and the overexpression phenotypes?

That is a reasonable assumption, however, for most of the observed phenotypes in the overexpressing plants, we did not detect a correlation between the level of expression of the transgene and the severity of the phenotype. This was in spite of the large amount of variation in levels of expression between transgenic plants - for instance, the CrLFY2 line with 31-fold OE compared to WT, exhibit comparable phenotypes to another closer to 2-fold. In most of our investigations, it appears that OE of CrLFY1 and CrLFY2 have the same phenotypic effects, with the exception of cell number in the gametophyte meristem. We believe it is more likely that this is due to functional divergence of the paralogs, as opposed to variation in OE levels, as CrLFY2 OE plants overexpressing 2-fold more than WT have the same meristem cell number as those overexpressing 31-fold. We have added lines [153-155] into the text highlighting that difference in expression does not correlate with differences in phenotype.

In Fig. 2, could you include how meristematic cells were defined (and marked by a dot in A-E) - 2:1 L/W ratio?

This is explained in the methods, in lines 608-611, and a short additional explanation has been added to the figure caption.

Typo in Line 131: CrLFY2: ***We could not find a typo here.***

Line 376: Lotus in this case refers to a legume, not an early-diverging angiosperm, please rephrase.

That is correct, and we have reassessed and removed the reference as it uninformative.

Line 462: could you clarify what you mean by "have clearly evolved"?

This was meant to be clarified in the next sentence; however, we have ultimately removed that section from the text.

Second decision letter

MS ID#: dev.204808R1

MS TITLE: LEAFY demonstrates functions in reproductive development of the gametophyte but not the sporophyte of the fern *Ceratopteris richardii*

AUTHORS: Hannah McConnell, Jancee R. Lanclos, Katelynn Willis, Nicholas Gjording, Genevieve Stockmann, Catalina Lind, Julin N. Maloof, Andrew R.G. Plackett and Verónica S. Di Stilio

Dear Dr Di Stilio,

I have now received all the referees reports on the above manuscript, and have reached a decision. The referees' comments are appended below, or you can access them online: please go to .

The overall evaluation is positive and we would like to publish a revised manuscript in Development, provided that the referees' comments can be satisfactorily addressed. At this point there are no new experiments required, but the conclusions need to be more circumspect in the ways reviewers 1 and 2 suggest. Please attend to all of the reviewers' comments in your revised manuscript and detail them in your point-by-point response. If you do not agree with any of their criticisms or suggestions explain clearly why this is so. If it would be helpful, you are welcome to contact us to discuss your revision in greater detail. Please send us a point-by-point response indicating your plans for addressing the referees' comments, and we will look over this and provide further guidance.

Reviewer 1

Advance summary and potential significance to field

Thanks for the additional explanation and data in this revised manuscript. The additional data adds considerably to the manuscript, and goes a long way to supporting the authors' conclusions. However, I am still not in favor of the title, and would suggest at least dropping the 'ancestral'. LFY expression in arabidopsis stamens and Marchantia antheridia suggests a shared ancient function in gametophytes, but this expression data alone is not sufficient to support shared function in arabidopsis, *Ceratopteris*, and *Marchantia*. In addition, while I am very glad the authors included the knock-down data in Fig. 4, these data do not firmly establish a sperm function for LFY, especially when combined with the data in Fig. 5, which shows no transmission defect. I understand that this lack of sperm release is classified as 'reproductive' in this manuscript, but really any trait can be classified as reproductive when discussing ephemeral gametophytes. Given all this uncertainty, I think it would be better/safer to change the title to something more accurate (e.g. 'LFY functions in embryonic and gametophyte development in *Ceratopteris* gametophytes'), or at least drop the ancestral in the title, to provide room for other possibilities that may be revealed with further experimentation.

Reviewer 2

Advance summary and potential significance to field

The revised manuscript has addressed several of my concerns, and includes new data intended to address two major concerns about 1) lack of LOF data to support the reported GOF phenotypes and 2) the conclusion that *Ceratopteris* LFY function supports an ancestral role for LFY in reproduction across vascular plants. I appreciate these additional efforts to further support their conclusions,

and indeed this data does provide new and interesting insights into CrLFY function. However, I still have concerns about the conclusions drawn, even considering this additional data. I detail these concerns, and provide a few other suggestions below. I do not think that additional data collection is likely to resolve these concerns, rather I recommend that a new "story" be considered to present this data. If a unifying story around the evolution of the angiosperm LFY reproductive role is considered most relevant, then I strongly recommend more circumspection and consideration of alternatives that are not ruled out by the current data.

Comments for the author

1. My primary concern is the conclusion highlighted in the title, namely that a CrLFY gametophyte reproductive role suggests an ancestral role for the vascular plants. In other sections it is suggested that the core reproductive angiosperm LFY function may have been recruited from this ancestral role. I remain skeptical of both conclusions. First, it is unlikely that the fern gametophyte meristem or sperm development activities of LFY are homologous to the reproductive function of LFY in angiosperms. There is no clear meristem in the highly reduced microgametophyte of angiosperms, nor any known LFY role in angiosperm sperm. It is possible that the transition to spore producing leaves in the sporophyte is homologous to the transition to flowering in angiosperms, which would indeed provide a basis for speculation about an ancestral role being modified or co-opted. However, the data suggests such a role is not shared across Ceratopteris and angiosperms. Second, even though the sperm and notch meristem functions proposed for CrLFY can be interpreted as "reproductive" (although see caveat below), there is currently no data to support that this role would be ancestral. At this point we only have data from a single fern lineage. Thus, it is just as likely that a gametophyte/sperm reproductive role is an apomorphy of Ceratopteris or the fern lineage more broadly. There is no evidence for a LFY role in gametophyte reproduction in bryophytes or other non-seed plant lineages. At this point, it is pre-mature to conclude that the gametophyte LFY phenotypes reveal an ancestral role. That is a possibility to be considered, but not a conclusion to be drawn.

2. The new data on notch meristem cell number of LFY RNAi knockdowns is inconclusive. I could not find the characterization of the lines shown in Figure S5 showing CrLFY1/2 levels (was this in a previous paper?). I agree that you would want to favor strong knock-down lines. However, even with this additional data, the fact that only 1 out of 7 lines shows a significant phenotype is worrisome. This could be a statistical anomaly, or a nonspecific result of the transgene insertion site. Without some duplication (i.e. multiple independent transgenic lines), this result is suggestive at best. Consequently, the reproducible GOF notch meristem phenotype does not yet support a role for CrLFY in the notch meristem.

3. I appreciate the new data provided about leaf number before sporing in Figure 1A. While there is no significant difference in the mean of these lines, there does appear to be more variance in vegetative fronds to sporing in lines where CrLFY is overexpressed. Is this apparent difference significant? Perhaps an F-test or a Bartlett's test could help? If there is indeed an increase in variance, this might point to a loss in developmental canalization for the transition to sporing when LFY is over-expressed, which would be worth reporting, even if hard to interpret given the GOF nature of the experiment.

Minor suggestions:

1. Abstract: "They evolved as a modification of the ancestral plant life cycle, whereby the haploid gamete-producing generation (gametophyte) became enclosed within the diploid, spore-producing (sporophyte)." This sentence would be true for seed plants more broadly, not for angiosperms in particular, where it was the seed that was enclosed by the carpel (Fruit), and the reproductive organs arranged in a single determinate branch (Flower).
2. Lines 251-253: Appears that the figure references should be to 4F, 4G, and 4H-I.
3. Lines 284-285: Is it possible that the presence of wt selfing sperm might explain some of the partial rescue of embryo development, when flooded with transgenic sperm?
4. Line 287: Crilly seems to be an auto-correct typo.
5. Line 400-403: I made this point in the previous review, but rescue of Arabidopsis by CrLFY implies nothing about the conservation of an ancestral function, it only reveals that the proteins share enough biochemical similarity that one can replace the other. A gene's developmental function depends on the network of evolved interactions present in the organism, which can only

be assessed in the organism proper. Thus it is possible to get rescue across species without a conserved ancestral function if the proteins are similar enough. Alternatively, the functions may be conserved but the protein sequences have diverged to such an extent that no rescue is seen across species.

Reviewer 3

Advance summary and potential significance to field

Thank you for addressing my comments, the revision looks good.

Second revision

Author response to reviewers' comments

Reviewer 1: Thanks for the additional explanation and data in this revised manuscript. The additional data adds considerably to the manuscript, and goes a long way to supporting the authors' conclusions. However, I am still not in favor of the title, and would suggest at least dropping the 'ancestral'. LFY expression in Arabidopsis stamens and Marchantia antheridia suggests a shared ancient function in gametophytes, but this expression data alone is not sufficient to support shared function in Arabidopsis, Ceratopteris, and Marchantia. In addition, while I am very glad the authors included the knock-down data in Fig. 4, these data do not firmly establish a sperm function for LFY, especially when combined with the data in Fig. 5, which shows no transmission defect. I understand that this lack of sperm release is classified as 'reproductive' in this manuscript, but really any trait can be classified as reproductive when discussing ephemeral gametophytes. Given all this uncertainty, I think it would be better/safer to change the title to something more accurate (e.g. 'LFY functions in embryonic and gametophyte development in Ceratopteris gametophytes'), or at least drop the ancestral in the title, to provide room for other possibilities that may be revealed with further experimentation.

We thank Reviewer 1 for their approval of the additional data included. In response to their continuing concerns about the use of the term ancestral we have changed the title to: "LEAFY demonstrates functions in reproductive development of the gametophyte but not the sporophyte of the fern Ceratopteris richardii."

Regarding their concerns about the expression data, while we think that this supports an ancestral male reproductive function for LFY, we agree that it is inconclusive. We have edited the manuscript's Discussion to clarify this point (lines 433-441).

We respectfully disagree with the reviewer's comment that 'any trait can be classified as reproductive when discussing ephemeral gametophytes'. Gametophytes of C. richardii do grow vegetatively and produce rhizoids, first via an apical cell and later from the notch meristem, as reviewed in Banks et al. 1999 (<https://doi.org/10.1146/annurev.arplant.50.1.163>), and under our growth conditions, gametophytes do not begin developing reproductive structures until 7 days after sowing, by which time a photosynthetic thallus is established. We also believe that it is reasonable to count the release of sperm from antheridia as a reproductive trait in the same way that a failure of anther dehiscence to release pollen is considered a reproductive failure in angiosperms. As such, we believe the data support keeping the reproductive aspect in the title.

Reviewer 2: SUMMARY OF THE ADVANCE MADE IN THIS PAPER AND ITS POTENTIAL SIGNIFICANCE TO THE FIELD

The revised manuscript has addressed several of my concerns, and includes new data intended

to address two major concerns about 1) lack of LOF data to support the reported GOF phenotypes and 2) the conclusion that Ceratopteris LFY function supports an ancestral role for LFY in reproduction across vascular plants. I appreciate these additional efforts to further support their conclusions, and indeed this data does provide new and interesting insights into CrLFY function. However, I still have concerns about the conclusions drawn, even considering this additional data. I detail these concerns, and provide a few other suggestions below. I do not think that additional data collection is likely to resolve these concerns, rather I recommend that a new "story" be considered to present this data. If a unifying story around the evolution of the angiosperm LFY reproductive role is considered most relevant, then I strongly recommend more circumspection and consideration of alternatives that are not ruled out by the current data.

We thank Reviewer 2 for their continued engagement in this process. We have endeavored to address their specific points (please see below). Regarding the paper's narrative, we continue to believe that the evolution of LFY's reproductive functions remains the most relevant and impactful; however, we have added text throughout to clarify the caveats and alternatives suggested. In addition to the edits described in the section below, we have revised the language of our conclusions to this effect in the abstract (line 35-37), Introduction (lines 141-146) and Discussion (lines 437-441, 492-507).

SUGGESTIONS TO AUTHORS

1. My primary concern is the conclusion highlighted in the title, namely that a CrLFY gametophyte reproductive role suggests an ancestral role for the vascular plants. In other sections it is suggested that the core reproductive angiosperm LFY function may have been recruited from this ancestral role. I remain skeptical of both conclusions. First, it is unlikely that the fern gametophyte meristem or sperm development activities of LFY are homologous to the reproductive function of LFY in angiosperms. There is no clear meristem in the highly reduced microgametophyte of angiosperms, nor any known LFY role in angiosperm sperm. It is possible that the transition to spore producing leaves in the sporophyte is homologous to the transition to flowering in angiosperms, which would indeed provide a basis for speculation about an ancestral role being modified or co-opted. However, the data suggests such a role is not shared across Ceratopteris and angiosperms. Second, even though the sperm and notch meristem functions proposed for CrLFY can be interpreted as "reproductive" (although see caveat below), there is currently no data to support that this role would be ancestral. At this point we only have data from a single fern lineage. Thus, it is just as likely that a gametophyte/sperm reproductive role is an apomorphy of Ceratopteris or the fern lineage more broadly. There is no evidence for a LFY role in gametophyte reproduction in bryophytes or other non-seed plant lineages. At this point, it is pre-mature to conclude that the gametophyte LFY phenotypes reveal an ancestral role. That is a possibility to be considered, but not a conclusion to be drawn.

We thank Reviewer 2 for their constructive skepticism. In light of their continuing concerns, we have edited the title to remove mention of ancestral functions. We have also changed the wording in the abstract (lines 35-37), introduction (lines 141-146) and discussion (lines 437-441) to clarify the strength of the available evidence and make it clear that we are not saying that an ancestral gametophyte function is proven, but a hypothesis. Furthermore, we note that the suggestion of co-option of LFY functions from the gametophyte was always phrased as a hypothesis, rather than a direct conclusion. We have edited the Discussion's conclusion to make that distinction unambiguous (lines 496-507). We believe it is important to include this as it represents the first specific and testable hypothesis regarding the evolutionary origin of that function.

Regarding their criticism about the lack of homology between the sporing transition and floral transition, we believe that there is, in fact, sufficient evidence in the literature to support developmental homology for this, even if there is no available genetic data. This is based on: 1) the homology of the shoot apex, which arose in the last common ancestor of vascular plants as supported by fossil evidence and comparative transcriptomics, 2) the fact that sporophyte reproduction in the form of sporogenesis is ancestral, predating even the vascular plants, and 3) that shoot apices in all vascular plant lineages undergo a development shift into a reproductive phase. The most parsimonious explanation is thus

that this reproductive transition is homologous. To clarify this point, we have included a summary in the Introduction that includes four new citations (Tomescu et al., 2014; Frank et al., 2015; Spencer et al., 2020; and Zhao et al., 2025) (Lines 78-84). In this context, we argue that our finding that LFY does not promote the sporing transition in Ceratopteris is not contradictory of this homology, but rather shows that at this point in vascular plant shoot evolution LFY does not yet have a genetic function in this transition and its apical role is purely vegetative- where it is expressed in reproductive apices it is presumably maintaining underlying apex function, rather than specific reproductive processes. This finding is significant because it suggests that the sporophytic LFY function in the MRCA of ferns and flowering plants was vegetative; thus, changes to LFY function enabling it to become part of flowering and floral development, evolved after this divergence, which was not previously known. The outcome of the research in this manuscript is that a detailed hypothesis regarding how that functional change may have occurred has been formulated.

We thank Reviewer 2 for highlighting their concerns about whether an ancestral gametophyte reproductive role can be inferred from our results in the context of other existing data. While we agree that there is not yet direct functional evidence for an ancestral gametophyte reproductive role, we make the following points:

1) LFY homolog expression is reported in the sperm-producing organs (antheridia) of the liverwort Marchantia polymorpha, similarly to Ceratopteris (as we already highlight in lines 428-437), providing indirect evidence for a possible ancestral function, particularly now that recent phylogenies have ascribed the bryophytes as a monophyletic clade (we added a citation here) and making Marchantia equally related to vascular plants as Physcomitrium (line 437-441).

2) There is also similar expression evidence for LFY expression in Arabidopsis pollen-producing organs (stamens), for which a function has not yet been ascribed. As such, we feel that the reviewer's comment that there is no evidence for an ancestral LFY function is not strictly supported, as LFY is expressed in association with male reproductive development across the land plants.

2. The new data on notch meristem cell number of LFY RNAi knockdowns is inconclusive. I could not find the characterization of the lines shown in Figure S5 showing CrLFY1/2 levels (was this in a previous paper?).

The RNAi transgenic lines were previously published and characterized, as cited in the text: Plackett, A., Conway, S.J., Hazelton, K.D.H., Rabbinoiwitsch, E.H., Langdale, J.A., Di Stilio, V.S., 2018. LEAFY maintains apical stem cell activity during shoot development in the fern Ceratopteris richardii. eLife. <https://doi.org/10.7554/eLife.39625>. We have edited the Results section to make this clearer (lines 200-202; 209-210).

I agree that you would want to favor strong knock-down lines. However, even with this additional data, the fact that only 1 out of 7 lines shows a significant phenotype is worrisome. This could be a statistical anomaly, or a nonspecific result of the transgene insertion site. Without some duplication (i.e. multiple independent transgenic lines), this result is suggestive at best.

Consequently, the reproducible GOF notch meristem phenotype does not yet support a role for CrLFY in the notch meristem.

We have reworded the relevant discussion section (lines, 407-409) to indicate that this result is suggestive and further edited the Discussion conclusion to reflect that the data on this point is not fully conclusive (starting at line 492).

3. I appreciate the new data provided about leaf number before sporing in Figure 1A. While there is no significant difference in the mean of these lines, there does appear to be more variance in vegetative fronds to sporing in lines where CrLFY is overexpressed. Is this apparent difference significant? Perhaps an F-test or a Bartlett's test could help? If there is indeed an increase in variance, this might point to a loss in developmental canalization for the transition to sporing when LFY is over-expressed, which would be worth reporting, even if hard to

interpret given the GOF nature of the experiment.

We appreciate the suggestion to investigate a potential increase in variance and have run an F-test. From this analysis, it does not appear that there is a significant increase in variance from WT. We have updated the Results section to include this (line 170).

Minor suggestions:

1. Abstract: "They evolved as a modification of the ancestral plant life cycle, whereby the haploid gamete-producing generation (gametophyte) became enclosed within the diploid, spore-producing (sporophyte)." This sentence would be true for seed plants more broadly, not for angiosperms in particular, where it was the seed that was enclosed by the carpel (Fruit), and the reproductive organs arranged in a single determinate branch (Flower).

Our statement is purposely broad; we assume that if it applies broadly to seed plants, it will also apply to angiosperms, because angiosperms are a type of seed plant. The existence of a carpel does not alter the fact that the haploid stage becomes just a few cells within the sporophyte. However, to address the reviewer's comment and avoid any perceived ambiguity, we have redrafted this sentence (lines 24-27) to clarify our intended meaning.

2. Lines 251-253: Appears that the figure references should be to 4F, 4G, and 4H-I.

Thank you for bringing this error to our attention; it has been corrected, and the missing panels are now included (lines 260-265).

3. Lines 284-285: Is it possible that the presence of wt selfing sperm might explain some of the partial rescue of embryo development, when flooded with transgenic sperm?

We thank Reviewer 2 for this reasonable alternative. Embryos that developed as a result of a WT female x 35S::CrLFY male cross were genotyped and were found to be transgenic. We have updated the Materials and Methods (lines 629-632) and Results sections (lines 289-291).

4. Line 287: Crilly seems to be an auto-correct typo.

Yes, thank you for noticing, it has been changed to CrLFY.

5. Line 400-403: I made this point in the previous review, but rescue of Arabidopsis by CrLFY implies nothing about the conservation of an ancestral function, it only reveals that the proteins share enough biochemical similarity that one can replace the other. A gene's developmental function depends on the network of evolved interactions present in the organism, which can only be assessed in the organism proper. Thus it is possible to get rescue across species without a conserved ancestral function if the proteins are similar enough. Alternatively, the functions may be conserved but the protein sequences have diverged to such an extent that no rescue is seen across species.

We thank Reviewer 2 for this point. While we agree that this is a valid technical criticism, we believe that this study is important to discuss in the context of our results. We have rephrased both our Introduction (lines 98-101) and Discussion, emphasizing the CrLFY2 protein's potential to partially perform the angiosperm function, provided the right genomic context is present. In response to the reviewer's concern over the interpretation of the results from this study, we have also included a statement in the Discussion clarifying that this is not by itself conclusive evidence of functional conservation (lines 419-422).

Reviewer 3: SUMMARY OF THE ADVANCE MADE IN THIS PAPER AND ITS POTENTIAL SIGNIFICANCE TO THE FIELD

Thank you for addressing my comments, the revision looks good.

Third decision letter

MS ID#: dev.204808R2

MS TITLE: LEAFY demonstrates functions in reproductive development of the gametophyte but not the sporophyte of the fern *Ceratopteris richardii*

AUTHORS: Hannah McConnell, Jancee R. Lanclos, Katelynn Willis, Nicholas Gjording, Genevieve Stockmann, Catalina Lind, Julin N. Maloof, Andrew R.G. Plackett and Verónica S. Di Stilio

Dear Dr Di Stilio,

I am happy to tell you that your manuscript has been accepted for publication in Development, pending our standard publication integrity checks.